# Bacterial colonization stimulates a complex physiological response in the immature human intestinal epithelium

**David R Hill[1], Sha Huang[1], Melinda S Nagy[1], Veda K Yadagiri[1], Courtney Fields[2], Dishari Mukherjee[3], Brooke Bons[2], Priya H Dedhia[4], Alana M Chin[1], Yu-Hwai Tsai[1], Shrikar Thodla[1], Thomas M Schmidt[3], Seth Walk[5], Vincent B Young[2†*], Jason R Spence[1,6†*]**

[1]Division of Gastroenterology, Department of Internal Medicine, University of Michigan, Ann Arbor, United States; [2]Division of Infectious Disease, Department of Internal Medicine, University of Michigan, Ann Arbor, United States; [3]Department of Microbiology and Immunology, University of Michigan, Ann Arbor, United States; [4]Department of Surgery, University of Michigan, Ann Arbor, United States; [5]Department of Microbiology and Immunology, Montana State University, Bozeman, United States; [6]Department of Cell andDevelopmental Biology, University of Michigan, Ann Arbor, United States

**\*For correspondence:**
youngvi@umich.edu (VBY);
spencejr@umich.edu (JRS)

[†]These authors contributed equally to this work

**Competing interests:** The authors declare that no competing interests exist.

**Abstract** The human gastrointestinal tract is immature at birth, yet must adapt to dramatic changes such as oral nutrition and microbial colonization. The confluence of these factors can lead to severe inflammatory disease in premature infants; however, investigating complex environment-host interactions is difficult due to limited access to immature human tissue. Here, we demonstrate that the epithelium of human pluripotent stem-cell-derived human intestinal organoids is globally similar to the immature human epithelium and we utilize HIOs to investigate complex host-microbe interactions in this naive epithelium. Our findings demonstrate that the immature epithelium is intrinsically capable of establishing a stable host-microbe symbiosis. Microbial colonization leads to complex contact and hypoxia driven responses resulting in increased antimicrobial peptide production, maturation of the mucus layer, and improved barrier function. These studies lay the groundwork for an improved mechanistic understanding of how colonization influences development of the immature human intestine.
DOI: https://doi.org/10.7554/eLife.29132.001

## Introduction

The epithelium of the gastrointestinal (GI) tract represents a large surface area for host-microbe interaction and mediates the balance between tolerance of mutualistic organisms and the exclusion of potential pathogens (*Peterson and Artis, 2014*). This is accomplished, in part, through the formation of a tight physical epithelial barrier, in addition to epithelial secretion of antimicrobial peptides and mucus (*Veereman-Wauters, 1996*; *Renz et al., 2011*). Development and maturation of the epithelial barrier coincides with the first exposure of the GI tract to microorganisms and the establishment of a microbial community within the gut (*Palmer et al., 2007*; *Koenig et al., 2011*). Although microorganisms have long been appreciated as the primary drivers of the postnatal expansion of adaptive immunity (*Renz et al., 2011*; *Shaw et al., 2010*; *Hviid et al., 2011*; *Abrahamsson et al., 2014*; *Arrieta et al., 2015*), and more recently as key stimuli in the development of digestion (*Erkosar et al., 2015*), metabolism (*Cho et al., 2012*), and neurocognitive function (*Diaz Heijtz*

**eLife digest** Human newborns are exposed to large numbers of bacteria at birth. They must transition from the protective, sterile environment of the womb into the bacteria-rich world. The gut, in particular, must adapt as bacteria colonize it. Many of the first bacteria found in the newborn gut form the basis of the bacterial communities needed for a healthy intestine throughout life. In some premature infants, bacterial colonization of the intestine may trigger harmful inflammation and a serious illness called necrotizing enterocolitis.

It is not known exactly how the immature intestine first responds to bacteria. It is also unclear what goes wrong that causes illness in some premature infants. Learning more about how a healthy newborn intestine becomes colonized and responds to this colonization may help medical professionals to better care for normal infants and those who are at risk of intestinal disease. But it has been difficult to study because there is not much newborn intestinal tissue available for scientific research. One solution would be to grow tissue in the laboratory that is like tissue found in the newborn intestine and see how it adapts to bacteria.

Now, Hill et al. use stem cells to grow a tissue in the laboratory that is very like immature newborn intestine and show that bacterial colonization helps it to mature. In the experiments, stem cells were grown into an intestine-like tissue. Analyses showed that this laboratory-grown tissue had the same patterns of gene expression as newborn intestines. Then, a type of bacteria called *Escherichia coli* that is normally found in the intestines of healthy babies was introduced to the intestine-like tissue. Hill et al. show that the initial contact with bacteria and changes in oxygen levels due to bacterial activity cause shifts in gene expression. These in turn stimulate the release of mucus and other protective responses.

A protein called NF-κB plays a central role in these normal bacteria-intestine interactions. Hill et al. show that using a drug to block NF-κB interferes with these processes. The experiments show that contact with bacteria encourages the immature intestine to protect itself from potential harm. More experiments like these may help scientists understand normal bacteria-intestine interactions in early life and how they may go wrong in disease. These studies might also help identify new treatments for babies with necrotizing enterocolitis.

DOI: https://doi.org/10.7554/eLife.29132.002

*et al., 2011*; *Clarke et al., 2014*; *Borre et al., 2014*; *Desbonnet et al., 2014*), it remains unclear how the human epithelial surface adapts to colonization and expansion of microorganisms within the immature GI tract.

Studies in gnotobiotic mice have improved our understanding of the importance of microbes in normal gut function since these mice exhibit profound developmental defects in the intestine (*Round and Mazmanian, 2009*; *Gensollen et al., 2016*; *Bry et al., 1996*; *Hooper et al., 1999*) including decreased epithelial turnover, impaired formation of microvilli (*Abrams et al., 1963*), and altered mucus glycosylation at the epithelial surface (*Bry et al., 1996*; *Goto et al., 2014*; *Cash et al., 2006*). However, evidence also suggests that the immature human intestine may differ significantly from the murine intestine, especially in the context of disease (*Nguyen et al., 2015*). For example, premature infants can develop necrotizing enterocolitis (NEC), an inflammatory disease with unknown causes. Recent reports suggest a multifactorial etiology by which immature intestinal barrier function predisposes the preterm infant to intestinal injury and inflammation following post-partum microbial colonization (*Neu and Walker, 2011*; *Morrow et al., 2013*; *Greenwood et al., 2014*; *Hackam et al., 2013*; *Afrazi et al., 2014*; *Fusunyan et al., 2001*; *Nanthakumar et al., 2011*). Rodent models of NEC have proven to be inadequate surrogates for studying human disease (*Tanner et al., 2015*). Therefore, direct studies of host-microbial interactions in the immature human intestine will be important to understand the complex interactions during bacterial colonization that lead to a normal gut development or disease.

Important ethical and practical considerations have limited research on the immature human intestine. For example, neonatal surgical specimens are often severely damaged by disease and not conducive for ex vivo studies. We and others have previously demonstrated that human pluripotent stem-cell-derived human intestinal organoids (HIOs) closely resemble immature intestinal tissue

(*Spence et al., 2011*; *Finkbeiner et al., 2015*; *Watson et al., 2014*; *Forster et al., 2014*; *Dedhia et al., 2016*; *Aurora and Spence, 2016*; *Chin et al., 2017*) and recent work has established gastrointestinal organoids as a powerful model of microbial pathogenesis at the mucosal interface (*Leslie et al., 2015*; *McCracken et al., 2014*; *Forbester et al., 2015*; *Hill and Spence, 2017*).

In the current work, we used HIOs as a model immature intestinal epithelium and a human-derived non-pathogenic strain of *E. coli* as a model intestinal colonizer to examine how host-microbe interactions affected intestinal maturation and function. Although the composition of the neonatal intestinal microbiome varies between individuals, organisms within the genera *Escherichia* are dominant early colonizers (*Gosalbes et al., 2013*; *Bäckhed et al., 2015*) and non-pathogenic *E. coli* are widely prevalent and highly abundant components of the neonatal stool microbiome (*Palmer et al., 2007*; *Koenig et al., 2011*; *Bäckhed et al., 2015*; *Morrow et al., 2013*). Microinjection of *E. coli* into the lumen of three-dimensional HIOs resulted in stable bacterial colonization in vitro, and using RNA-sequencing, we monitored the global transcriptional changes in response to colonization. We observed widespread, time-dependent transcriptional responses that are the result of both bacterial contact and luminal hypoxia resulting from bacterial colonization in the HIO. Bacterial association with the immature epithelium increased antimicrobial defenses and resulted in enhanced epithelial barrier function and integrity. We observed that NF-κB is a central downstream mediator of the transcriptional changes induced by both bacterial contact and hypoxia. We further probed the bacterial contact and hypoxia-dependent epithelial responses using experimental hypoxia and pharmacological NF-κB inhibition, which allowed us to delineate which of the transcriptional and functional responses of the immature epithelium were oxygen and/or NF-κB dependent. We found that NF-κB-dependent microbe-epithelial interactions were beneficial by enhancing barrier function and protecting the epithelium from damage by inflammatory cytokines. Collectively, these studies shed light on how microbial contact with the immature human intestinal epithelium can lead to modified function.

## Results

### Pluripotent stem-cell-derived intestinal epithelium transcriptionally resembles the immature human intestinal epithelium

Previous work has demonstrated that stem-cell-derived human intestinal organoids resemble immature human duodenum (*Watson et al., 2014*; *Finkbeiner et al., 2015*; *Tsai et al., 2017*). Moreover, transplantation into immunocompromised mice results in HIO maturation to an adult-like state (*Watson et al., 2014*; *Finkbeiner et al., 2015*). These analyses compared HIOs consisting of epithelium and mesenchyme to whole-thickness human intestinal tissue, which also possessed cellular constituents lacking in HIOs such as neurons, blood vessels and immune cells (*Finkbeiner et al., 2015*). Thus, the extent to which the HIO epithelium resembles immature/fetal intestinal epithelium remained unclear. To address this gap and further characterize the HIO epithelium relative to fetal and adult duodenal epithelium, we isolated and cultured epithelium from HIOs grown entirely in vitro, from fetal duodenum, adult duodenum, or HIOs that had been transplanted into the kidney capsule of NSG immuno-deficient mice and matured for 10 weeks. These epithelium-only derived organoids were expanded in vitro in uniform tissue culture conditions for 4–5 passages and processed for RNA-sequencing (RNA-seq) (*Figure 1—figure supplement 1*). Comparison of global transcriptomes between all samples in addition to human embryonic stem cells (hESCs) used to generate HIOs (*Finkbeiner et al., 2015*; E-MTAB-3158) revealed a clear hierarchy in which both in vitro grown HIO epithelium ($p=5.06 \times 10^{-9}$) and transplanted epithelium ($p=7.79 \times 10^{-14}$) shares a substantially greater degree of similarity to fetal small intestinal epithelium (*Figure 1—figure supplement 1A*).

While unbiased clustering demonstrated that transplanted epithelium closely resembles fetal epithelium, we noted a shift toward the adult transcriptome that resulted in a relative increase in the correlation between transplanted HIO epithelium and adult duodenum-derived epithelium grown in vitro (*Figure 1—figure supplement 1B*, $p=1.17 \times 10^{-4}$). Principle component analysis (PCA) of this multi-dimensional gene expression dataset (*Figure 1—figure supplement 1C*) corroborated the correlation analysis, and indicated that PC1 was correlated with developmental stage (PC1, 27.75% cumulative variance) and PC2 was correlated with tissue maturation status (PC2, 21.49% cumulative

variance); cumulatively, PC1 and PC2 accounted for 49.24% of the cumulative variance between samples, suggesting that developmental stage and tissue maturation status are major sources of the transcriptional variation between samples. HIO epithelium clustered with fetal epithelium along PC2, whereas transplanted HIO epithelium clustered with adult epithelium.

We further used differential expression analysis to demonstrate that in vitro grown HIO epithelium is similar to the immature human intestine, whereas in vivo transplanted HIO epithelium is similar to the adult epithelium. To do this, we identified differentially expressed genes through two independent comparisons: (1) human fetal vs. adult epithelium; (2) HIO epithelium vs. transplanted HIO epithelium. Genes enriched in transplanted HIO epithelium relative to the HIO epithelium were compared to genes enriched in the adult duodenum relative to fetal duodenum (*Figure 1—figure supplement 1D*). There was a highly significant correlation between $\log_2$-transformed expression ratios where transplanted HIOs and adult epithelium shared enriched genes while HIO and fetal epithelium shared enriched genes (p=2.6 $\times$ $10^{-28}$). This analysis supports previously published data indicating that the epithelium from HIOs grown in vitro recapitulates the gene expression signature of the immature duodenum and demonstrates that the HIO epithelium is capable of adopting a transcriptional signature that more strongly resembles adult duodenum following transplantation into mice.

## HIOs can be stably associated with non-pathogenic *E. coli*

Given that the HIO epithelium recapitulates many of the features of the immature intestinal epithelium, we set out to evaluate the effect of bacterial colonization on the naïve HIO epithelium. Previous studies have established that pluripotent stem-cell-derived intestinal organoids can be injected with live viral (*Finkbeiner et al., 2012*) or bacterial pathogens (*Leslie et al., 2015*; *Engevik et al., 2015*; *Forbester et al., 2015*); however, it was not known if HIOs could be stably co-cultured with non-pathogenic microorganisms. We co-cultured HIOs with the non-motile human-derived *Esherichia coli* strain ECOR2 (*Ochman and Selander, 1984*). Whole genome sequencing and phylogentic analysis demonstrated that *E. coli* str. ECOR2 is closely related to other non-pathogenic human *E. coli* and only distantly related to pathogenic *E. coli* and *Shigella* isolates (*Figure 1—figure supplement 3*). We developed a microinjection technique to introduce live *E. coli* into the HIO lumen in a manner that prevented contamination of the surrounding media (*Figure 1—figure supplement 2*). HIOs microinjected with $10^5$ live *E. coli* constitutively expressing GFP exhibit robust green fluorescence within 3 hr of microinjection (*Figure 1A* and *Video 1*). Numerous *E. coli* localized to the luminal space at 48 hr post-microinjection and are present adjacent to the HIO epithelium, with some apparently residing in close opposition to the apical epithelial surface (*Figure 1B*).

In order to determine the minimum number of colony-forming units (CFU) of *E. coli* required to establish short term colonization (24 hr), we microinjected increasing numbers of live *E. coli* suspended in PBS into single HIOs and collected and determined the number of bacteria in the luminal contents at 24 hr post-microinjection (*Figure 1C*). Single HIOs can be stably colonized by as few as 5 CFU *E. coli* per HIO with 77.8% success (positive luminal culture and negative external media culture at 24 hr post-injection) and 100% success at ≥100 CFU per HIO (*Figure 1C*). Increasing the number of CFU *E. coli* microinjected into each HIO at $t = 0$ did result in a significant increase in the mean luminal CFU per HIO at 24 hr post-microinjection at any dose (ANOVA p=0.37; *Figure 1C*). Thus, the 24 hr growth rate of *E. coli* within the HIO lumen $\left( \frac{CFU \times HIO^{-1}_{t=24}}{CFU \times HIO^{-1}_{t=0}} \right)$ was negatively correlated with the CFU injected ($r^2$ = 0.625, p=3.1 $\times$ $10^{-12}$; *Figure 1C*).

Next, we examined the stability of HIO and *E. coli* co-cultures over time in vitro. HIOs were microinjected with 10 CFU *E. coli* and maintained for 24–72 hr (*Figure 1D*). Rapid expansion of *E. coli* density within the HIO lumen was observed in the first 24 hr, with relatively stable bacterial density at 48–72 hr. A 6.25-fold increase in bacterial density was observed between 24 and 72 hr post-microinjection (p=0.036). Importantly, samples taken from the external HIO culture media were negative for *E. coli* growth.

Finally, we examined the stability of HIO cultures following *E. coli* microinjection (*Figure 1E*). A total of 48 individual HIOs were microinjected with $10^4$ CFU *E. coli* each. Controls were microinjected with sterile PBS alone. We found that external culture media was sterile in 100% of control HIOs throughout the entire experiment, and in 100% of *E. coli* injected HIOs on days 0–2 post-microinjection. On days 3–9 post-microinjection some cultured media was positive for *E. coli* growth;

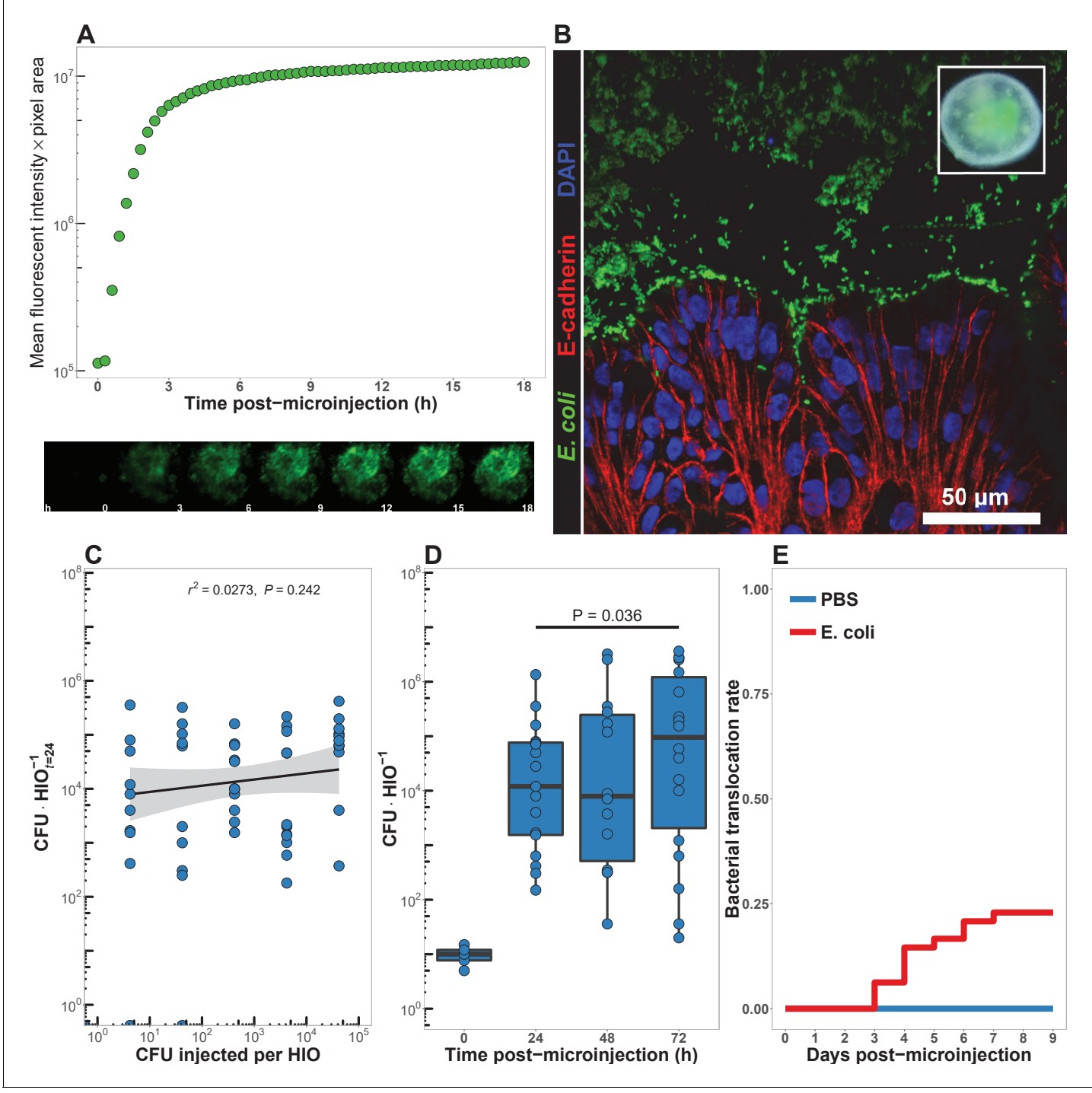

**Figure 1.** HIOs can be stably associated with non-pathogenic E. coli. (**A**) Mean fluorescent intensity of a human intestinal organoid (HIO) containing live GFP+ *E. coli* str. ECOR2. The lower panels show representative images from the time series. Representative of three independent experiments. *Video 1* is an animation corresponding to this dataset. (**B**) Confocal micrograph of the HIO epithelium (E-cadherin) in direct association with GFP+ *E. coli* at 48 hr post-microinjection with $10^4$ live *E. coli*. 60X magnification. (**C**) Luminal CFU per HIO *E. coli* at 24 hr post-microinjection relative to the injected concentration of $5 \times 10^{-1}$ to $5 \times 10^5$ CFU per HIO at the start of the experiment. N = 10 biological replicates per *E. coli* dose. The $r^2$ and P value shown in the figure represent the results of a linear regression analysis of the relationship between the 24 hr change in CFU/HIO and the initial number of CFU injected. (**D**) Luminal CFU per HIO at 0–72 hr following microinjection with 10 CFU *E. coli* per HIO. N = 13–17 replicate HIOs per time point. The p-value represents the results of a two-tailed Student's *t*-test comparing the two conditions indicated. (**E**) Daily proportion of HIO cultures with no culturable *E. coli* in the external media following *E. coli* microinjection (N = 48) or PBS microinjection (N = 8).

DOI: https://doi.org/10.7554/eLife.29132.003

*Figure 1 continued on next page*

*Figure 1 continued*

The following figure supplements are available for figure 1:

**Figure supplement 1.** Pluripotent stem-cell-derived intestinal epithelium transcriptionally resembles the immature human intestinal epithelium.
DOI: https://doi.org/10.7554/eLife.29132.004
**Figure supplement 2.** Phylogenetic tree based on maximum liklihood genomic distance among *E.coli* str.
DOI: https://doi.org/10.7554/eLife.29132.005
**Figure supplement 3.** HIO colonization protocol.
DOI: https://doi.org/10.7554/eLife.29132.006

however, 77.08% of *E. coli* injected HIOs were negative for *E. coli* in the external culture media throughout the timecourse. Additional control experiments were conducted to determine if the HIO growth media had any effect on *E. coli* growth. *E.coli*-inoculated HIO growth media showed that the media itself allowed for robust bacterial growth, and therefore the absence of *E. coli* growth in external media from HIO cultures could not be attributed to the media composition alone (*Figure 1—figure supplement 3*). Thus, the large majority of *E. coli* colonized HIOs remain stable for an extended period when cultured in vitro and without antibiotics.

## Bacterial colonization elicits a broad-scale, time-dependent transcriptional response

Colonization of the immature gut by microbes is associated with functional maturation in both model systems (*Kremer et al., 2013*; *Sommer et al., 2015*; *Broderick et al., 2014*; *Erkosar et al., 2015*) and in human infants (*Renz et al., 2011*). To evaluate if exposing HIOs to *E. coli* led to maturation at the epithelial interface, we evaluated the transcriptional events following microinjection of live *E. coli* into the HIO lumen. PBS-injected HIOs (controls) and HIOs co-cultured with *E. coli* were collected for transcriptional analysis after 24, 48 and 96 hr (*Figure 2*). At 24 hr post-microinjection, a total of 2018 genes were differentially expressed (adjusted-FDR < 0.05), and the total number of differentially expressed genes was further increased at 48 and 96 hr post-microinjection relative to PBS-injected controls (*Figure 2A*). Principle component analysis demonstrated that global transcriptional activity in HIOs is significantly altered by exposure to *E. coli*, with the degree of transcriptional change relative to control HIOs increasing over time (*Figure 2B*).

Gene set enrichment analysis (GSEA) (*Subramanian et al., 2005*) using the GO (*Ashburner et al., 2000*; *Gene Ontology Consortium, 2015*) and REACTOME (*Croft et al., 2014*; *Fabregat et al., 2016*) databases to evaluate RNA-seq expression data revealed coordinated changes in gene expression related to innate anti-microbial defense, epithelial barrier production, adaptation to low oxygen, and tissue maturation (*Figure 2C*). Innate antimicrobial defense pathways, including genes related to NF-κB signaling, cytokine production, and Toll-like receptor (TLR) signaling were strongly upregulated at 24 hr post-microinjection and generally exhibited decreased activation at later time points. GSEA also revealed changes in gene expression consistent with reduced oxygen levels or hypoxia, including the induction of pro-angiogenesis signals. A number of pathways related to glycoprotein synthesis and modification, including O-linked mucins, glycosaminoglycans, and proteoglycans, were up-regulated in the initial stages of the transcriptional response

**Video 1.** Animation of individual epifluorescent microscopy images from a human intestinal organoid (HIO) containing live GFP$^+$ *E. coli* str. ECOR2. Images were captured at 10 min intervals over the course of 18 hr an coalated in sequential order. Representative of three independent experiments. See *Figure 1A*.
DOI: https://doi.org/10.7554/eLife.29132.007

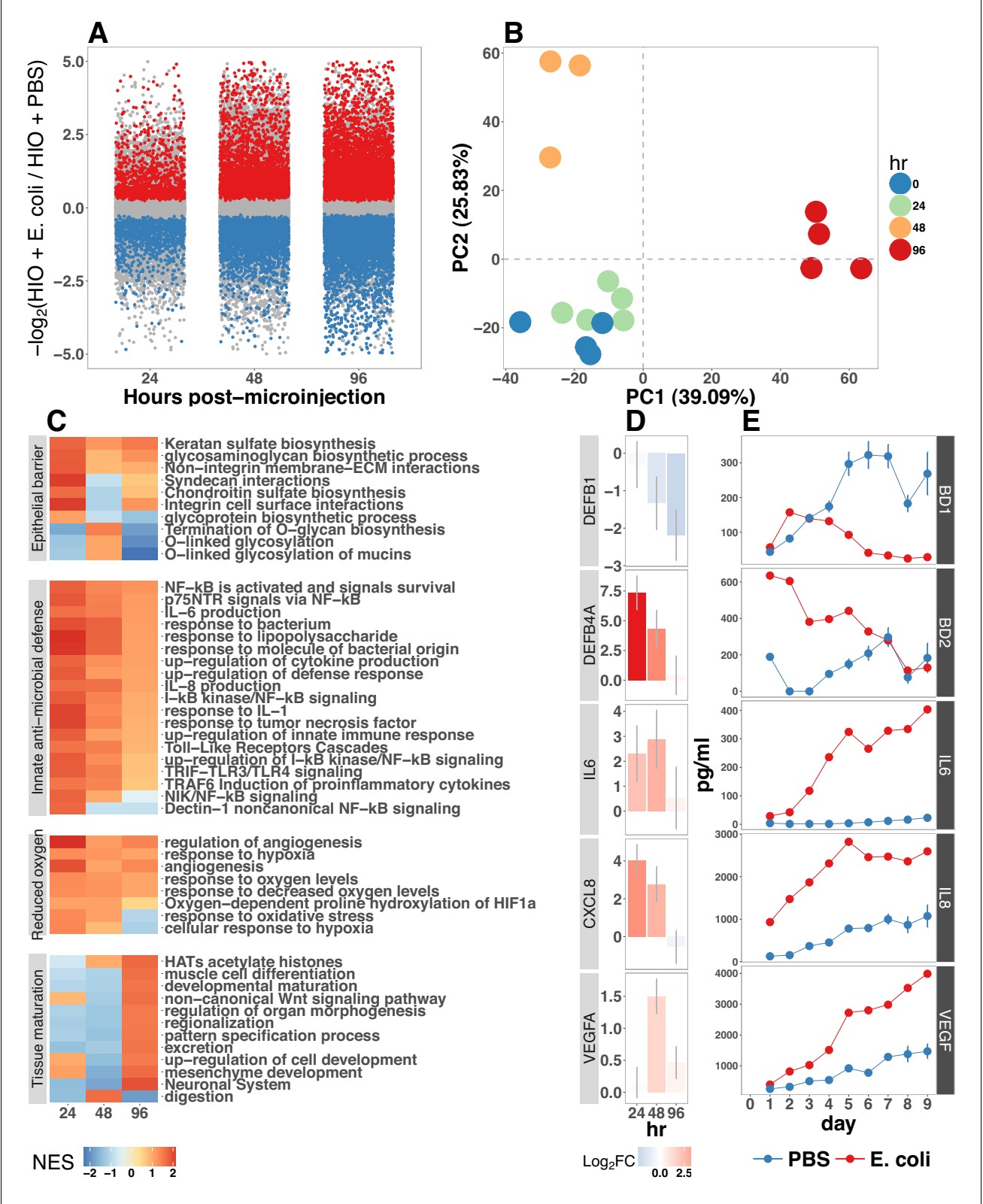

**Figure 2.** (A) Log2-transformed fold change in normalized RNA-seq gene counts in *E. coli* colonized HIOs at 24, 48, and 96 hr post-microinjection with 10⁴ live *E. coli* relative to PBS-injected HIOs. Differentially expressed genes (FDR-adjusted p-value < 0.05) are indicated in red (up-regulated) or blue (down-regulated). Plotted results are the mean fold change per gene for each group. (B) Principle component plot of HIOs at 0–96 hr post-microinjection derrived from whole-transcriptome RNA-seq normalized gene counts. Cumulative explained variance for PC1 and PC2 is indicated as a

*Figure 2 continued on next page*

*Figure 2 continued*

percentage on the x- and y-axes, respectively. (**C**) Heat map of normalized enrichment scores (NES) from GSEA of normalized RNA-seq expression data using the GO and REACTOME databases. A positive value of NES indicates activation of a given gene set and a negative value suggests relative suppression of a gene set. All NES scores are calculated relative to PBS-microinjected controls. (**D**) Mean $\log_2$ fold change in normalized RNA-seq gene counts at 24–96 hr post microinjection relative to PBS-injected control HIOs. (**E**) Protein secretion at 0–9 days post-microinjection with PBS or *E. coli* as measured by ELISA in the supernatant of HIO cultures. The genes given in D correspond to the proteins measured in E. $N = 4$ (0 hr), 5 (24 hr), 3 (48 hr), and 4 (96 hr) biological replicates consisting of 5–6 pooled HIOs per replicate for panels A-D. $N = 48$ *E. coli*-injected HIOs and $N = 8$ PBS-injected HIOs for panel E.

DOI: https://doi.org/10.7554/eLife.29132.008

(Syndecans, integrins), exhibited a somewhat delayed onset (O-linked mucins), or exhibited consistent activation at all time points post-microinjection (Keratan sulfate and glycosaminoglycan biosynthesis). Finally, genes sets associated with a range of processes involved in tissue maturation and development followed a distinct late-onset pattern of expression. This included broad gene ontology terms for organ morphogenesis, developmental maturation, and regionalization as well as more specific processes such as differentiation of mesenchymal and muscle cells, and processes associated with the nervous system (*Figure 2C*).

We also made correlations between upregulated genes in the RNA-seq data (*Figure 2D*) and protein factors present in the organoid culture media following *E. coli* microinjection (*Figure 2E*). β-defensin 1 (*DEFB1* (gene); BD-1 (protein)) and β-defensin 2 (*DEFB4A* (gene); BD-2 (protein)) exhibited distinct patterns of expression, with both *DEFB1* and its protein product BD-1 stable at 24 hr after *E. coli* microinjection but relatively suppressed at later time points, and *DEFB4A* and BD-2 strongly induced at early time points and subsiding over time relative to PBS-injected controls. By contrast, inflammatory regulators IL-6 and IL-8 and the pro-angiogenesis factor VEGF were strongly induced at the transcriptional level within 24–48 hr of *E. coli* microinjection. Secretion of IL-6, IL-8, and VEGF increased over time, peaking at 5–9 days after *E. coli* association relative to PBS-injected controls (*Figure 2E*). Taken together, this data demonstrates a broad-scale and time-dependent transcriptional response to *E. coli* association with distinct early- and late-phase patterns of gene expression and protein secretion.

## Bacterial colonization results in a transient increase in epithelial proliferation and the maturation of enterocytes

While the transcriptional analysis demonstrated strong time-dependent changes in the cells that comprise the HIO following *E. coli* colonization, we hypothesized that exposure to bacteria may also alter the cellular behavior and/or composition of the HIO. Previous studies have demonstrated that bacterial colonization promotes epithelial proliferation in model organisms (*Bates et al., 2006*; *Cheesman et al., 2011*; *Neal et al., 2013*; *Kremer et al., 2013*; *Ijssennagger et al., 2015*). We examined epithelial proliferation in HIOs over a timecourse of 96 hr by treating HIOs with a single 2 hr exposure of 10 $\mu$M EdU added to the culture media from 22 to 24 hr after microinjection with $10^4$ CFU *E. coli* or PBS alone. HIOs were subsequently collected for immunohistochemistry at 24, 48, and 96 hr post-microinjection (*Figure 3*). The number of proliferating epithelial cells (Edu$^+$ and E-cadherin$^+$) was elevated by as much as three-fold in *E. coli*-colonized HIOs relative to PBS-treated HIOs at 24 hr (*Figure 3A–B*). However, at 48 hr post-microinjection, the proportion of EdU + epithelial cells was significantly decreased in *E. coli* colonized HIOs relative to control treated HIOs. This observation was supported by another proliferation marker, KI67 (*Gerdes et al., 1984*) (*Figure 3B*), as well as RNA-seq data demonstrating an overall suppression of cell cycle genes in *E. coli* colonized HIOs relative to PBS-injected HIOs at 48 hr post-microinjection (*Figure 3—figure supplement 1*). By 96 hr post-microinjection the proportion of EdU+ epithelial cells was nearly identical in *E. coli* and PBS-treated HIOs (*Figure 3B*). Collectively, these results suggest that *E. coli* colonization is associated with a rapid burst of epithelial proliferation, but that relatively few of the resulting daughter cells are retained subsequently within the epithelium.

The transcription factor Sox9 is expressed by progenitor cells in the murine intestinal epithelium (*Bastide et al., 2007*; *Mori-Akiyama et al., 2007*), and several epithelial subtypes are derived from a Sox9-expressing progenitor population in the mature intestinal epithelium (*Bastide et al., 2007*;

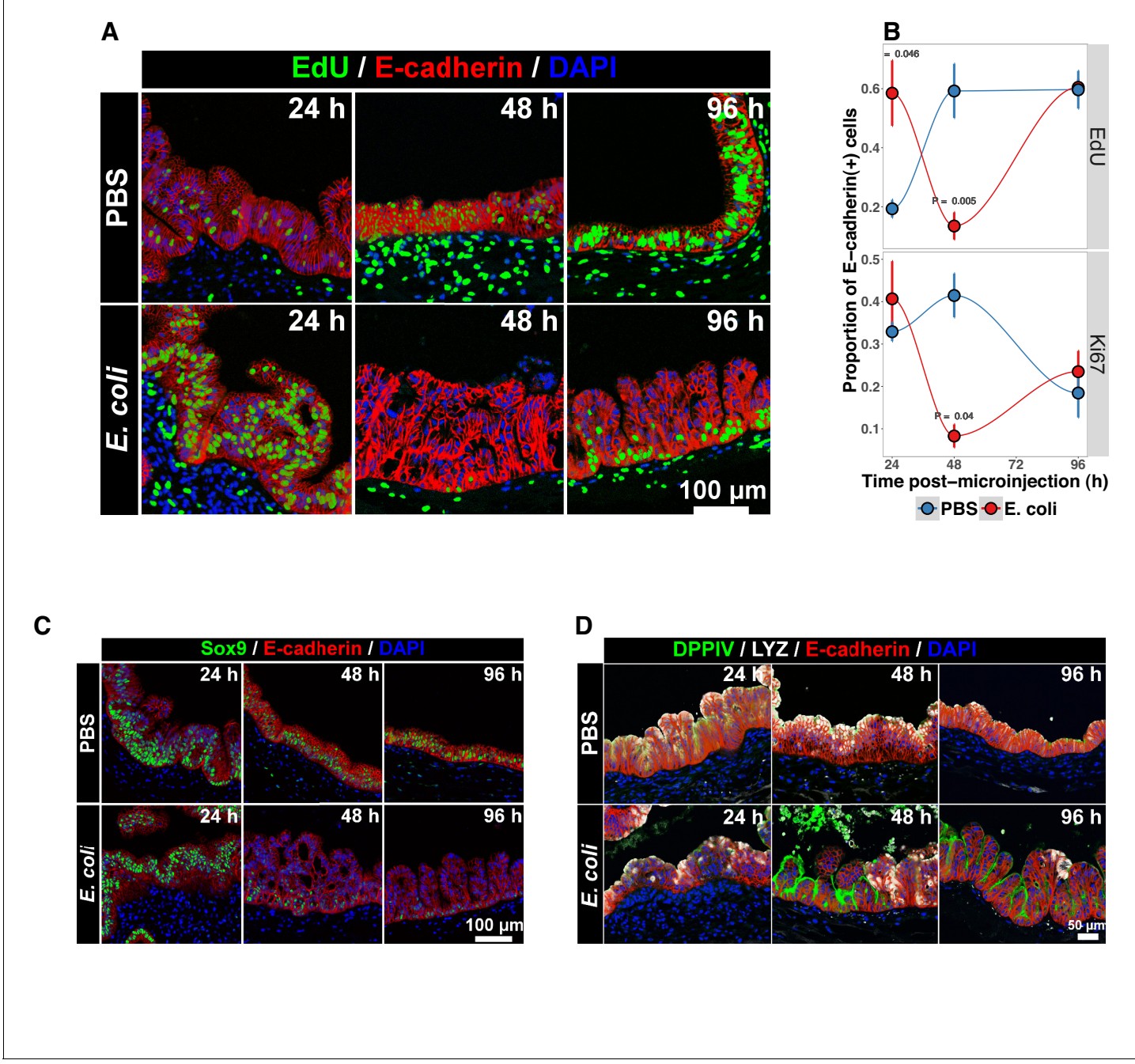

**Figure 3.** Bacterial colonization results in a transient increase in epithelial proliferation and the maturation of enterocytes. (**A**) Representative confocal micrographs of HIOs injected with PBS or $10^4$ CFU *E. coli* str. ECOR2 at 24–96 hr post-microinjection and stained with fluorescent indicators for for EdU+DNA, E-cadherin, or nuclei (DAPI) as indicated in the figure labels . All HIOs were exposed to 10 $\mu$M EdU at 22 hr post-microinjection and EdU was removed at 24 hr. Panels are representative of 4 HIOs per timepoint per treatment condition. (**B**) Quantification of the number of EdU-positive and Ki67-positive epithelial cells (E-cadherin+ cells) per 10X confocal microscopy field. One 10X confocal microscopy field consisting of 200–1000 epithelial cells was collected from each of 4 HIOs per timepoint per treatment group. The error bars represent the standard error of the mean and the p-values reflect the results of an unpaired two-tailed Student's *t*-test comparing the PBS-injected HIOs to the *E. coli*-injected HIOs at that timepoint. (**C**) Representative confocal micrographs of HIOs injected with PBS or $10^4$ CFU *E. coli* str. ECOR2 at 24–96 hr post-microinjection and stained with fluorescent antibodies for Sox9, E-cadherin, or nuclei (DAPI) as indicated in the figure labels. Panels are representative of 4 HIOs per timepoint per treatment condition.

DOI: https://doi.org/10.7554/eLife.29132.009

The following figure supplement is available for figure 3:

*Figure 3 continued on next page*

Figure 3 continued

**Figure supplement 1.** Pathview (*Luo and Brouwer, 2013*) plot of the KEGG (*Kanehisa and Goto, 2000*) pathway showing the Cell mitotic cycle ('HSA 04110') superimposed with RNA-seq expression data corresponding to the $\log_2$-transformed fold change in expression of cell cycle transcripts from HIOs microinjected with *E. coli* relative to PBS-injected HIOs at 48 h post-microinjection.
DOI: https://doi.org/10.7554/eLife.29132.010

*Furuyama et al., 2011*). We examined SOX9 expression in HIOs following microinjection with *E. coli* or PBS alone over a 96 hr time course (*Figure 3C*). In the PBS-treated HIOs, the majority of epithelial cells exhibited robust nuclear SOX9 expression at all time points examined. However, SOX9 expression was dramatically reduced in *E. coli*-colonized HIOs at 48–96 hr after microinjection and was notably distributed in nuclei farthest from the lumen and adjacent to the underlying mesenchyme, mirroring the altered distribution of EdU + nuclei seen in *Figure 3B*. This observation suggests that there is a reduction in the number of progenitor cells in the HIO epithelium following *E. coli* colonization and implies that other epithelial types may account for a greater proportion of the HIO epithelium at later time points post-colonization. We saw no appreciable staining for epithelial cells expressing goblet, Paneth, or enteroendocrine cell markers (MUC2, DEFA5, and CHGA, respectively; negative data not shown). However, expression of the small intestinal brush border enzyme dipeptidyl peptidase-4 (DPPIV) was found to be robustly expressed in the *E. coli*-colonized HIOs at 48 and 96 hr post-microinjection (*Figure 3D*). DPPIV was not detected in any of the PBS-injected HIOs at any timepoint. Lysozyme (LYZ), an antimicrobial enzyme expressed by Paneth-like progenitors in the small intestinal crypts *Bevins and Salzman (2011)*, was widely distributed throughout the epithelium of PBS-treated HIOs as we have previously described (*Spence et al., 2011*) (*Figure 3D*). However, in *E. coli*-colonized HIOs, LYZ expression was restricted to distinct clusters of epithelial cells and, notably, never overlapped with DPPIV staining (*Figure 3D*). Given that *bona fide* Paneth Cell markers (i.e. DEFA5) were not observed in any HIOs, it is likely that the LYZ expression is marking a progenitor-like population of cells. Taken together, these experiments indicate that *E. coli* colonization induces a substantial but transient increase in the rate of epithelial proliferation followed by a reduction and redistribution of proliferating epithelial progenitors and differentiation of a population of cells expressing small intestinal enterocyte brush boarder enzymes over a period of 2–4 days.

## *E. coli* colonization is associated with a reduction in luminal $O_2$

The mature intestinal epithelium is characterized by a steep oxygen gradient, ranging from 8% oxygen within the bowel wall to <2% oxygen in the lumen of the small intestine (*Fisher et al., 2013*). Reduction of oxygen content in the intestinal lumen occurs during the immediate perinatal period (*Gruette et al., 1965*), resulting in changes in epithelial physiology (*Glover et al., 2016*; *Kelly et al., 2015*; *Colgan et al., 2013*; *Zeitouni et al., 2016*) that helps to shape the subsequent composition of the microbiota (*Schmidt and Kao, 2014*; *Espey, 2013*; *Albenberg et al., 2014*; *Palmer et al., 2007*; *Koenig et al., 2011*). Analysis of the global transcriptional response to *E. coli* association in the immature intestinal tissue revealed pronounced and coordinated changes in gene expression consistent with the onset of hypoxia (*Figure 2C–E*). We therefore measured oxygen concentration in the lumen of control HIOs and following microinjection of live *E. coli* using a 50 $\mu$m diameter fiber-optic optode (*Figure 4A–B*). Baseline oxygen concentration in the organoid lumen was 8.04 ± 0.48%, which was significantly reduced relative to the external culture media (18.86 ± 0.37%, p=3.6 × $10^{-11}$). At 24 and 48 hr post-microinjection, luminal oxygen concentration was significantly reduced in *E. coli*-injected HIOs relative to PBS-injected HIOs (p=0.04 and p=5.2 × $10^{-05}$, respectively) reaching concentrations as low as 1.67 ± 0.62% at 48 hr (*Figure 4A*). *E. coli* injected HIOs were collected and CFU were enumerated from luminal contents at 24 and 48 hr post-microinjection. We observed a highly significant negative correlation between luminal CFU and luminal oxygen concentration where increased density of luminal bacteria was correlated with lower oxygen concentrations ($r_2$ = 0.842, p=6.86 × $10^{-5}$; *Figure 4B*). Finally, in order to assess relative oxygenation in the epithelium itself, we utilized a small molecule pimonidazole (PMDZ), which forms covalent conjugates with thiol groups on cytoplasmic proteins only under low-oxygen conditions (*Arteel et al., 1998*). Fluorescent immunochemistry demonstrated enhanced PMDZ uptake in *E. coli* associated HIO epithelium, and in HIOs grown in 1% $O_2$ as a positive control when compared to to PBS-injected

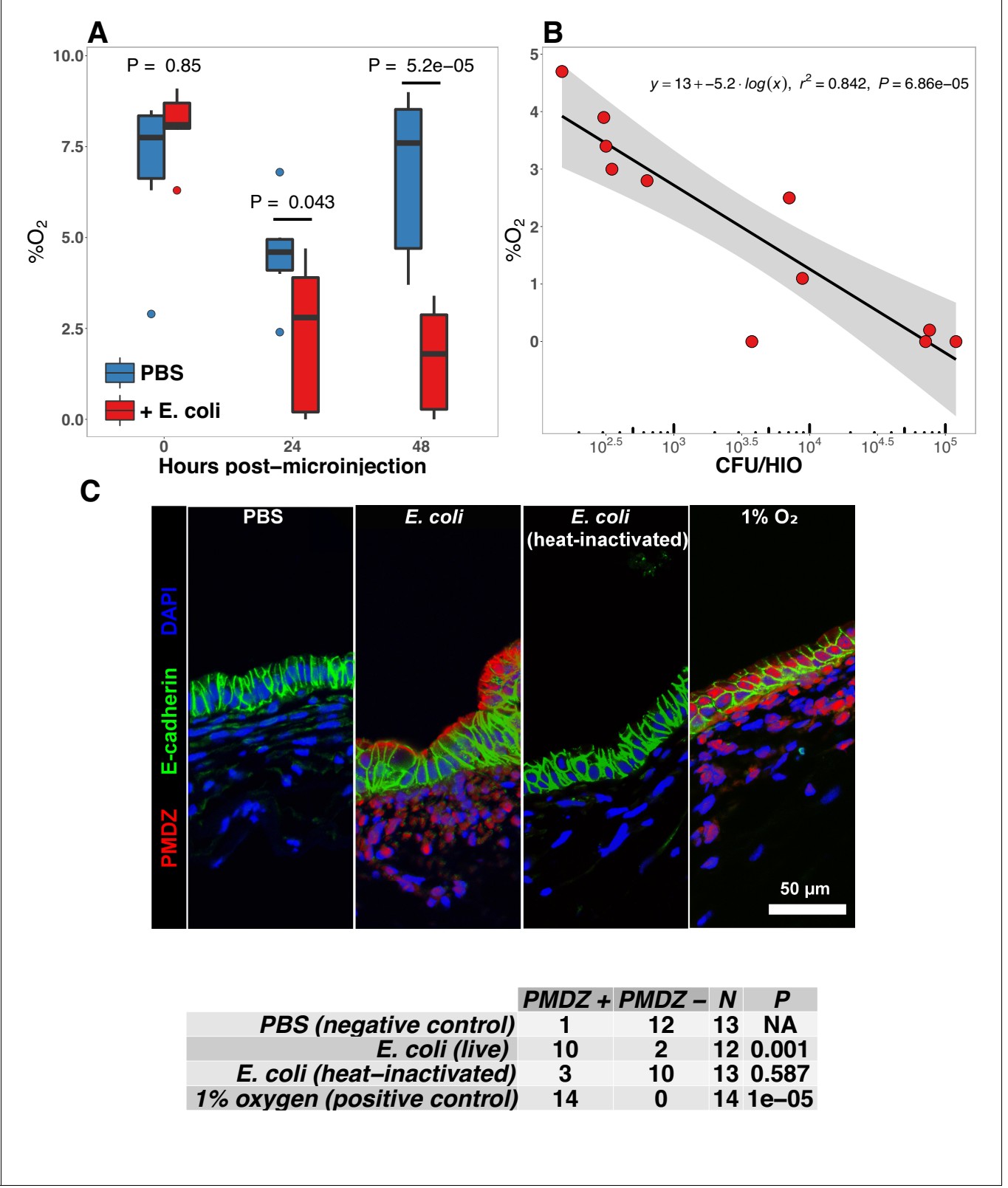

**Figure 4.** E. coli colonization is associated with a reduction in luminal oxygen concentration. (A) Luminal oxygen concentration in human intestinal organoids at 0–48 hr post-microinjection with $10^4$ CFU live E. coli. p Values reflect results of unpaired one-tailed Students t-tests for the comparisons indicated. N = 6–11 replicate HIOs per treatment group per time point. (B) Linear regression analysis of luminal CFU E. coli per organoid at and luminal oxygen concentration in the same organoid 24 hr post-microinjection . (C) Confocal micrographs of the HIO epithelium in PBS- and E. coli-injected

*Figure 4 continued on next page*

*Figure 4 continued*

HIOs at 48 hr post-microinjection. Images are representative of the replicates detailed in the table, with 12–14 replicate HIOs per treatment group pooled from two separate experiments. Individual HIOs were scored as PMDZ$^+$ or PMDZ$^-$ based on the presence or absence, respectively, of PMDZ conjugates as detected by immunofluorescent microscopy. p-Values represent the results of $\chi^2$ contingency tests comparing the distribution of PMDZ$^+$ and PMDZ$^-$ HIOs in the PBS-treated group to each of the other conditions.

DOI: https://doi.org/10.7554/eLife.29132.011

HIOs, or HIOs injected with heat killed *E. coli* at 48 hr post-microinjection (*Figure 4C*). Thus, luminal and epithelial oxygen is reduced following microinjection of *E. coli* into the HIO, consistent with data in mice showing that the in vivo epithelium is in a similar low-oxygen state in normal physiological conditions (*Schmidt and Kao, 2014*; *Kelly et al., 2015*; *Kim et al., 2017*).

## NF-κB integrates complex microbial and hypoxic stimuli

*E. coli* colonization elicits a robust transcriptional response in immature intestinal tissue (*Figure 2*) that is associated with the onset of luminal oxygen depletion and relative tissue hypoxia (*Figure 4*). We set out to determine whether we could assign portions of the transcriptional response to direct interaction with microbes or to the subsequent depletion of luminal oxygen. In the RNA-seq analysis (*Figure 2*), NF-κB signaling emerged as a major pathway involved in this complex host-microbe interaction, and NF-κB has been shown by others to act as a transcriptional mediator of both microbial contact and the response to tissue hypoxia (*Rius et al., 2008*; *Gilmore, 2006*; *Wullaert et al., 2011*). Gene Ontology and REACTOME pathway analysis showed that NF-κB signaling components are also highly up-regulated following microinjection of *E. coli* into HIOs (*Figure 2C* and *Figure 5—figure supplement 1A*). Thus, we assessed the role of NF-κB signaling in the microbial contact-associated transcriptional response and the hypoxia-associated response using the highly selective IKKβ inhibitor SC-514 (*Kishore et al., 2003*; *Litvak et al., 2009*) to inhibit phosphorylation and activation of the transcription factor p65 (*Figure 5—figure supplement 1B*). Another set of HIOs was simultaneously transferred to a hypoxic chamber and cultured in 1% O$_2$ with and without SC-514. At 24 hr post-treatment, HIOs were harvested for RNA isolation and RNA-seq. We devised an experimental scheme that allowed us to parse out the relative contributions of microbial contact and microbe-associated luminal hypoxia in the transcriptional response to association with live *E. coli* (*Figure 5A* and *Figure 5—figure supplement 1C*). First, we identified a set of genes significantly up-regulated (log$_2$FC > 0 and FDR-adjusted p-value < 0.05) by microinjection of either live *E. coli* or heat-inactivated *E. coli* (contact dependent genes). From this gene set, we identified a subset that was suppressed by the presence of NF-κB inhibitor SC-514 during association with either live or heat-inactivated *E. coli* (log$_2$FC < 0 and FDR-adjusted p-value < 0.05; Gene Set I, *Figure 5B*). Thus, Gene Set I represents the NF-κB dependent transcriptional response to live or dead *E. coli*. Genes induced by live or heat-inactivated *E. coli* but not suppressed by SC-514 were considered NF-κB independent (Gene Set III, *Figure 5B*). Likewise, we compared genes commonly up-regulated by association with live *E. coli* and those up-regulated under 1% O$_2$ culture conditions. A subset of genes induced by either live *E. coli* or 1% O$_2$ culture but suppressed by the presence of NF-κB inhibitor was identified as the NF-κB-dependent hypoxia-associated transcriptional response (Gene Set II, *Figure 5B*). Genes induced by live *E. coli* or hypoxia but not inhibited by the presence of NF-κB inhibitor were considered NF-κB independent transcriptional responses to microbe-associated hypoxia (Gene Set IV). Gene lists for each gene set are found in *Supplementary file 1*.

Following the identification of these four gene sets, we then applied over-representation analysis using the GO and REACTOME pathway databases to identify enriched pathways for each of the four gene sets, resulting in four clearly distinguishable patterns of gene pathway enrichment (*Figure 5C*). Contact with either live or heat-inactivated *E. coli* is sufficient to promote expression of genes involved in maintaining epithelial barrier integrity and mucin production, an effect that is suppressed in the presence of NF-κB inhibitor. Additionally, key developmental pathways including epithelial morphogenesis, digestive tract development, and expression of digestive enzymes appear to be driven primarily by bacterial association and are largely NF-κB dependent. Robust innate and adaptive defense requires both bacterial contact and hypoxia, with some genes associated with antigen processing and cytokine signaling being NF-κB dependent (Gene Set II) and others associated with

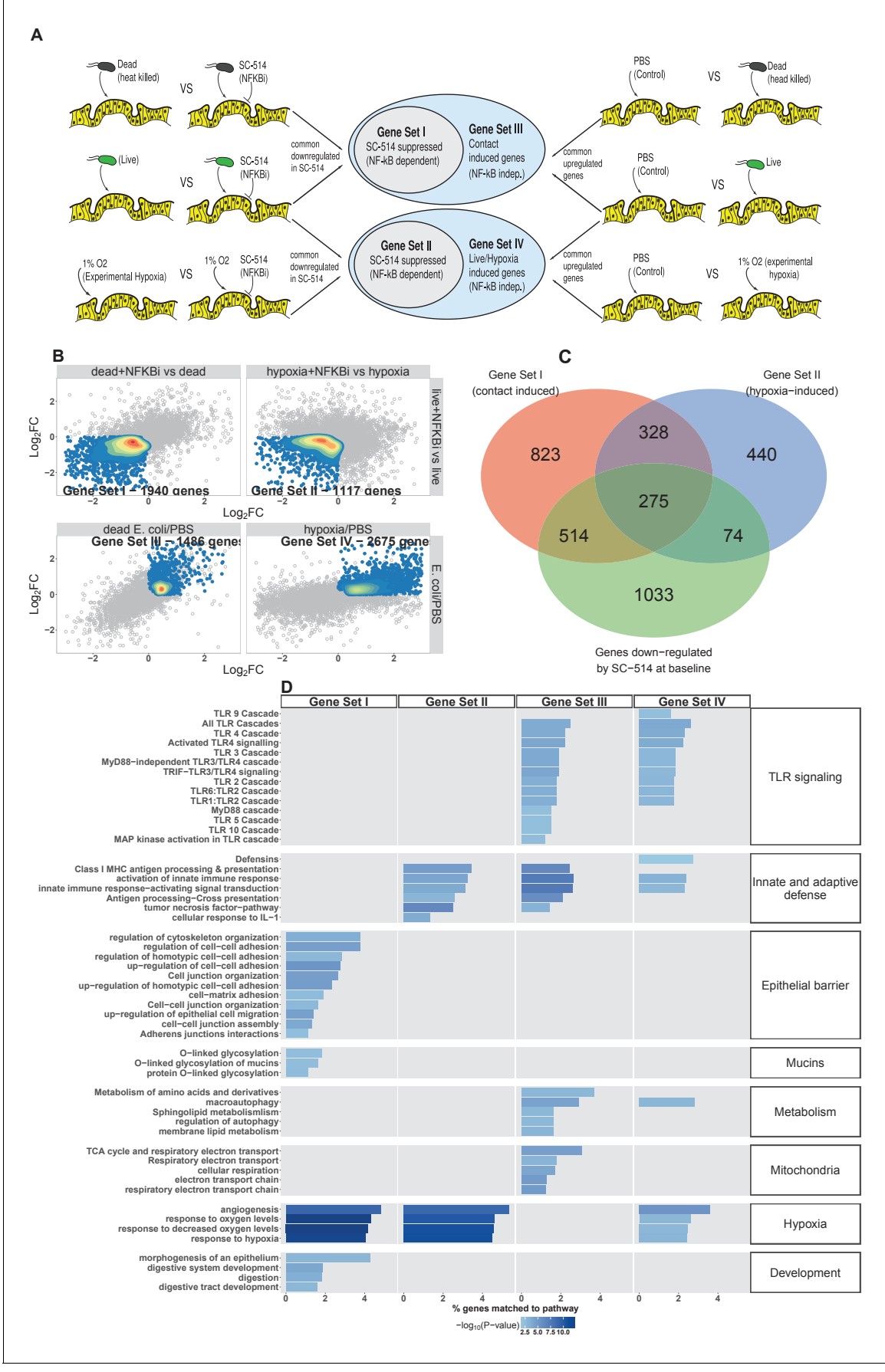

**Figure 5.** NF-κB integrates complex microbial and hypoxic stimuli. (**A**) Analysis scheme for identifying genes sets representing the components of the transcriptional response to live *E. coli* that could be recapitulated with heat-inactivated *E. coli* (contact induced) or hypoxia (microbial-associated hypoxia induced) as well as the subsets of genes induced through NF-κB dependent signaling. HIOs were microinjected with PBS, $10^4$ CFU *E. coli* or an equivalent concentration of heat-inactivated *E. coli* and cultured under standard cell culture conditions or hypoxic conditions (1% $O_2$, 5% $CO_2$, 94% $N_2$) with and without 10 μM SC-514. (**B**) Scatter plots with density overlay indicating the genes meeting the *a priori* criteria identified in panel A with an FDR-adjusted p-value of < 0.05 for the comparisons listed on the axes of the plot. (**C**) Bar plot of the proportion of genes in the input gene sets mapping to each pathway from the GO and REACTOME databases enrichment p-values for each of the gene sets identified in A. Pathways with enrichment p-values > 0.01 were excluded from the plot. Results represent *N* = 4–5 biological replicates per treatment condition, with each replicate consisting of 5–6 pooled and identically treated HIOs.

DOI: https://doi.org/10.7554/eLife.29132.012

The following figure supplements are available for figure 5:

**Figure supplement 1.** NF-κB signaling pathway acticvation in HIOs.

DOI: https://doi.org/10.7554/eLife.29132.013

**Figure supplement 2.** Secretion of AMPs (BD-1 & BD-2), cytokines (IL-6 & IL-8), and thepro-angiogenesis growth factor VEGF in HIOs microinjected with PBS, 10 4 CFU *E.*

DOI: https://doi.org/10.7554/eLife.29132.014

NF-κB-independent gene sets (Gene Sets III and IV). Genes associated with antimicrobial defensin peptides were enriched only in the hypoxia-asociated, NF-κB-independent gene set (Gene Set IV), suggesting that antimicrobial peptides are regulated by mechanisms that are distinct from other aspects of epithelial barrier integrity such as mucins and epithelial junctions (Gene Set I). TLR signaling components were is broadly enhanced by live *E. coli* and associated with both microbial contact and hypoxia were largely NF-κB independent (Gene Sets III and IV). There was a notable transcriptional signature suggesting metabolic and mitochondrial adaptation to bacteria that was independent of NF-κB and primarily driven by bacterial contact rather than hypoxia (Gene Set III).

To interrogate the transcriptional changes influenced by SC-514 exposure, we examined over-represented genes sets from the GO and REACTOME databases in genes that were significantly up- or down-regulated by treatment with SC-514 alone (*Figure 5—figure supplement 1C and D*) . Notably, SC-514 alone does not appear to have a strong effect on the pathways identified in *Figure 5C* as key NF-κB-dependent responses to bacterial contact and/or hypoxia. In *Figure 5—figure supplement 1E*, we examined the degree of overlap between Gene Set I, Gene Set II, and the set of genes that are significantly down-regulated in PBS-injected HIOs treated with SC-514. This analysis demonstrates that the majority of genes in Set I and Set II are not significantly down-regulated in PBS-injected HIOs treated with SC-514. The most significant effects of SC-514 alone among Gene Set I and Gene Set II genes are related to metabolism, redox state, and ribosomal dynamics (*Figure 5—figure supplement 1F*). Thus, the effect of SC-514 alone cannot account for the NF-κB-dependent changes in innate and adaptive defense, epithelial barrier integrity, angiogenesis and hypoxia signaling, or intestinal development following bacterial contact and/or hypoxia during colonization.

Finally, we also examined the role of microbial contact and hypoxia in colonization-induced changes in AMP, cytokine, and growth factor secretion using ELISA (*Figure 5—figure supplement 2*). Consistent with findings from the RNA-seq data, these results indicate that there are diverse responses to bacterial contact and hypoxia. We observed cases where cytokines were induced by either microbial contact or hypoxia alone (IL-6), other cases where hypoxia appeared to be the dominant stimuli (BD-1), and a third regulatory paradigm in which the response to live *E. coli* evidently results from the cumulative influence of bacterial contact and hypoxia (BD-2, IL-8, VEGF). Taken together, this analysis demonstrates that association of immature intestinal epithelium with live *E. coli* results in a complex interplay between microbial contact and microbe-associated hypoxia-induced gene expression and protein secretion.

## Bacterial colonization promotes secretion of antimicrobial peptides

Antimicrobial peptides (AMPs) are key effectors for innate defense of epithelial surfaces (*Muniz et al., 2012*) and act to inhibit microbial growth through direct lysis of the bacterial cell wall and modulation of bacterial metabolism (*Ganz, 2003*; *Bevins and Salzman, 2011*; *O'Neil and O'Neil, 2003*; *Vora et al., 2004*; *Brogden, 2005*). Defensin gene expression is highly up-regulated

following microinjection of *E. coli* into HIOs (*Figures 2D–E and* and *4C*). Using an annotated database of known AMPs (*Wang et al., 2016*) to query our RNA-seq datasets, we found that several AMPs are up-regulated in the immature intestinal epithelium following *E. coli* association (*Figure 6A*). Among these, DEFB4A and DEFB4B, duplicate genes encoding the peptide human β-defensin 2 (*Harder et al., 1997*), were the most highly up-regulated; other AMPs induced by *E. coli* association included multi-functional peptides CCL20, CXCL2, CXCL1, CXCL6, CXCL3, REG3A (*Cash et al., 2006*), and LTF (*Figure 6A*). Analysis of RNA-seq data from HIOs microinjected with live or heat-killed *E .coli* with and without NF-κB inhibitor or culture of HIOs under hypoxic conditions had indicated that defensin genes were enriched among the set of NF-κB-independent genes induced by hypoxia (*Figure 5C*). We examined *DEFB4A* expression specifically (*Figure 6B*) and found that relative to control treatment, microinjection of live *E. coli* resulted in a 7.38-fold increase in normalized *DEFB4A* expression. Consistent with the notion that *DEFB4A* expression is induced by hypoxia and is not dependent on NF-κB signaling, NF-κB inhibitor treated HIOs injected with *E. coli* still showed an ~8-fold increase in gene expression and hypoxia-cultured HIOs showed a ~5.5-fold

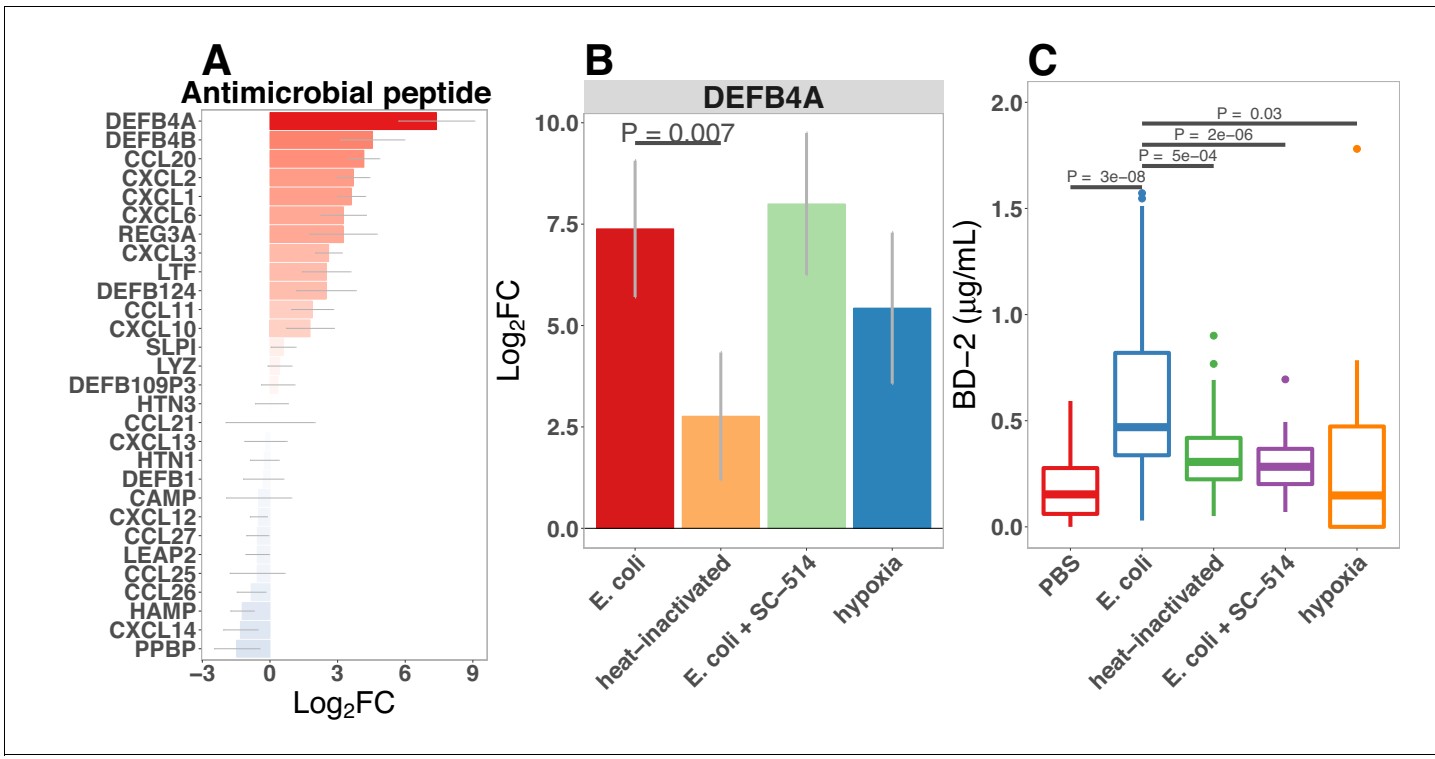

**Figure 6.** Bacterial colonization promotes secretion of antimicrobial peptides. (A) Normalized fold change in antimicrobial peptide (AMP) gene expression in *E. coli*-associated HIOs at 24 hr relative to PBS control treatment. (B) Normalized fold change in expression of DEFB4A, the gene encoding human β-defensin 2 (BD-2) peptide, in each of the conditions indicated relative to PBS control treatment. Results in panels A and B represent *N* = 4–5 biological replicates per treatment condition, with each replicate consisting of 5–6 pooled and identically treated HIOs. (C) Concentration of BD-2 peptide in culture supernatant at 24 hr as measured by ELISA in HIO cultures treated as indicated. *N* = 10–14 individually treated HIOs per treatment condition with data combined from three independant replicate experiments. (D) Optical density (600 nm) of *E. coli* suspension cultures supplemented with PBS or BD-2 at 10-min intervals over an 18 hr period at 37. (E) Carrying capacity (K) of media supplemented with varying concentrations of BD-2 derrived from the growth curves presented in panel D. *N* = 8 biological replicates per treatment group for panels D and E. p-Values represents the results of a two-tailed Student's *t*-test for the comparisons indicated.

DOI: https://doi.org/10.7554/eLife.29132.015

The following figure supplements are available for figure 6:

**Figure supplement 1.** Normalized fold change in expression of DEFB4B, a duplicated gene encoding human β-defensin 2 (BD-2) peptide, in each of the conditions indicated relative to PBS control treatment.
DOI: https://doi.org/10.7554/eLife.29132.016

**Figure supplement 2.** BD-2 inhibits E. coli growth in vitro.
DOI: https://doi.org/10.7554/eLife.29132.017

induction (*Figure 6B*). On the other hand, microinjection with heat-inactivated *E. coli* resulted in *DEFB4A* induction that was significantly lower relative to microinjection with live *E. coli* (p=0.007. A similar pattern of expression was observed for *DEFB4B* (*Figure 6—figure supplement 1*).

We also examined secretion of human β-defensin 2 peptide (BD-2) in the supernatant of *E. coli*-associated HIOs (*Figure 2E* and *Figure 6C*). BD-2 secretion was increased 3.4-fold at 24 hr following *E. coli* microinjection (p=$2.7 \times 10^{-8}$). However, heat-inactivation of *E. coli* or addition of NF-κB inhibitor resulted in suppression of BD-2 secretion relative to live *E. coli* (p=0.00051 and $1.6 \times 10^{-6}$, respectively).

To determine if the levels of BD-2 produced by HIOs and secreted into the media were sufficient to retard bacterial growth, we tested the effect of BD-2 at concentrations recapitulating the baseline state in the HIO (~0.1 μg/ml) and following microinjection with *E.coli* (~1 μg/ml) on in vitro growth of *E. coli* over 18 hr (*Figure 6D*). Although there was little effect on *E. coli* density during initial log-phase growth, BD-2 reduced the amount of time bacteria spent in log-phase growth, and *E. coli* density was significantly decreased over time in bacterial growth media supplemented BD-2 (p=0.001), resulting in a significant decrease in the effective in vitro carrying capacity, or maximum population density (*Figure 6E*, p=$8 \times 10^{-4}$). Furthermore, concentrations of BD-2 consistent with those found in HIO/E. coli supernatant (1 μg/ml) was significantly more inhibitory than low concentration BD-2 (0.1 μg/ml) in our in vitro growth assay (p=0.013). Additional data suggest that the inhibitory activity of BD-2 in vitro is not specific to *E. coli* str. ECOR2 and is dependent upon maintenance of BD-2 protein structure, since BD-2 similarly inhibited growth of *E. coli* str. K12, and heat-inactivated BD-2 lost these inhibitory effects (*Figure 6—figure supplement 2*). From this set of experiments, we conclude that *E. coli* colonization promotes enhanced expression of AMPs, including BD-2, at concentrations that are sufficient to suppress microbial growth.

## Bacterial colonization promotes expression of epithelial Mucins and glycotransferases

Mucins are an essential component of epithelial integrity, serving as a formidable barrier to microbial invasion and repository for secreted AMPs (*Bergstrom and Xia, 2013*; *Cornick et al., 2015*; *Johansson and Hansson, 2016*; *Kim and Ho, 2010*). Mucin synthesis requires a complex series of post-translational modifications that add high-molecular-weight carbohydrate side chains to the core mucin protein (*Varki, 2017*). Our RNA-seq data suggested that mucin gene expression is dependent on both bacterial contact and NF-κB signaling (*Figure 5C*). Therefore, we examined expression of genes in control and *E. coli* microinjected HIOs that encode mucin core proteins as well as the glycotransferases that generate the wide variety of post-translational mucin modifications (*Figure 7A*). Although some glycotransferases were increased at 24 hr after *E. coli* microinjection, expression of mucin core proteins and many glycotransferases reached peak levels at 48 hr after the introduction of *E. coli* to the HIO lumen (*Figure 7A*). Periodic Acid-Schiff and Alcian blue staining (PAS/AB) of sections taken from HIOs at 48 hr after *E. coli* microinjection reveal the formation of a robust mucin layer at the apical epithelial surface consisting of both acidic (AB-positive) and neutral (PAS-positive) glycoprotein components, suggesting a rich matrix of O-linked mucins, glycosaminoglycans, and proteoglycans (*Figure 7B–C*). Interestingly, we observed that *E. coli* association caused an initial induction of *MUC5AC* at 48 hr that was reduced by 96 hr (*Figure 7A*). *MUC5AC* is most highly expressed within the gastric mucosa but has also been reported in the duodenal epithelium (*Buisine et al., 1998*, *2001*; *Rodríguez-Piñeiro et al., 2013*). On the other hand, *MUC2* is more commonly associated with the duodenum, and increased more slowly, showing peak expression after 96 hr of association with *E. coli* (*Figure 7A*). Co-staining of control HIOs and *E. coli* microinjected HIOs demonstrated colocalization with *Ulex europaeus* agglutinin I (UEA1), a lectin with high specificity for the terminal fucose moiety Fucα1-2Gal-R (*Figure 7D*). This suggests that following *E. coli* association, HIOs produce mucins with carbohydrate modifications associated with bacterial colonization in vivo (*Cash et al., 2006*; *Hooper et al., 1999*; *Goto et al., 2014*).

RNA-seq data suggested that O-linked mucins were highly enriched among the subset of genes induced by bacterial contact in an NF-κB-dependent manner (*Figure 5*). We examined this phenomenon at the level of individual glycosyltransferase and mucin genes (*Figure 7E*). *E. coli* induced transcription of mucins and glycosyltransferases (*Figure 7E*) and mucin secretion (*Figure 7—figure supplement 1*) was suppressed in the presence of NF-κB inhibitor SC-514. Furthermore, culture of HIOs under hypoxia conditions was not sufficient to promote transcription of genes involved in

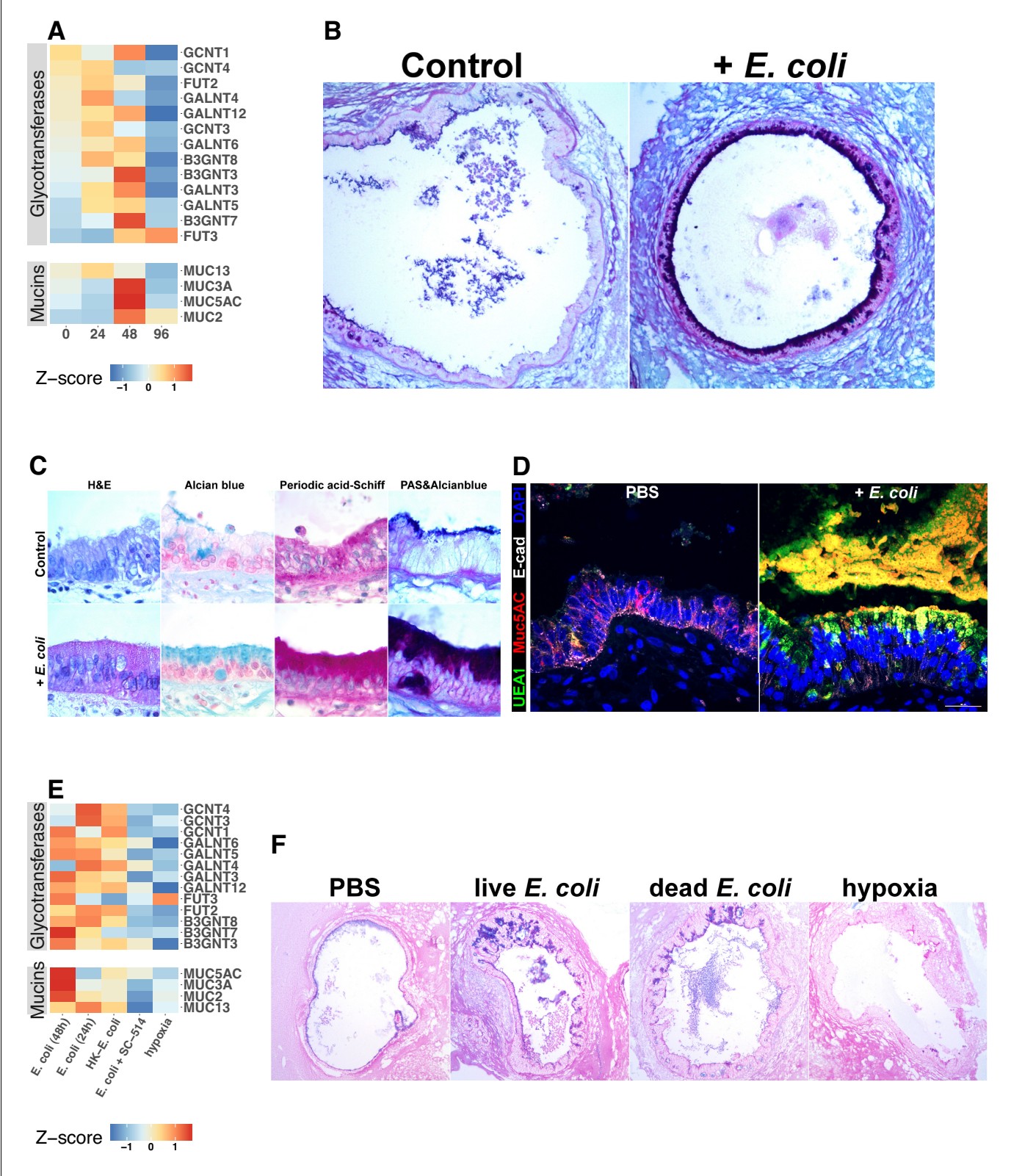

**Figure 7.** Bacterial colonization promotes expression of epithelial mucins and glycotransferases. (**A**) Heatmap of normalized RNA-seq glycotransferase and mucin gene counts of HIOs associated with *E.coli* at 0–96 hr post-microinjection. $N = 4$ (0 hr), 5 (24 hr), 3 (48 hr), and 4 (96 hr) biological replicates consisting of 5–6 pooled HIOs per replicate. (**B**) Periodic acid-Schiff and Alcian Blue (PAS-AB) staining of control HIOs or HIOs microinjected with *E. coli* and cultured for 48 hr at 10X magnification. (**C**) HIO epithelium from control HIOs or HIOs microinjected with *E. coli* and cultured for 48 hr stained with

*Figure 7 continued on next page*

*Figure 7 continued*

H and E, AB, PAS, or PAS-AB and imaged under 100X light microscopy. (**D**) Confocal micrograph of HIO epithelium from a control HIO or an HIO microinjected with *E. coli* and cultured for 48 hr. Nuclei are stained blue with DAPI, and fluorescent antibody-labeled proteins E-cadherein and Mucin 5 AC are pseudocolored in white or red, respectively. UEA1 lectin is used to label the carbohydrate moiety Fucα1-2Gal-R, which is pseudo colored in green. 60X optical magnification. (**E**) Heatmap of normalized RNA-seq glycotransferase and mucin gene counts of HIOs associated with live or heat-inactivated *E. coli*, *E. coli* + NF-κB inhibitor (SC-514) or HIOs cultured under hypoxic conditions for 24 hr. Results represent the mean of N = 4–5 biological replicates per treatment condition, with each replicate consisting of 5–6 pooled and identically treated HIOs. (**F**) PAS-AB staining of HIOs treated as indicated in the figure labels for 24 hr. 10X magnification. Histological and immunofluorescent images in panels B-D and F are representative of three or more independent experiments, each consisting of 5–10 HIOs per treatment group.
DOI: https://doi.org/10.7554/eLife.29132.018
The following figure supplement is available for figure 7:

**Figure supplement 1.** Representative confocal micrographs of HIOs treated as indicated.
DOI: https://doi.org/10.7554/eLife.29132.019

mucin synthesis (*Figure 7E*). This result was confirmed with PAS/AB staining of HIOs microinjected with PBS, live or heat-inactivated *E. coli*, or cultured under hypoxic conditions for 24 hr, where bacterial contact promoted formation of a mucus layer while PBS microinjection or culture under hypoxic conditions did not (*Figure 7F*). Taken together, these results indicate that association of the immature intestinal epithelium with *E. coli* promotes robust mucus secretion through an NF-κB-dependent mechanism and that hypoxia alone is not sufficient to recapitulate *E. coli*-induced mucus production.

## NF-κB signaling is required for the maintenance of barrier integrity following bacterial colonization

Having established that the immature intestinal epithelium in HIOs (*Figure 1—figure supplement 1*) can be stably associated with non-pathogenic *E. coli* (*Figure 1*), resulting in broad changes in transcriptional activity (*Figure 2*) and leading to elevated production of AMPs (*Figure 6*) and epithelial mucus secretion (*Figure 7*), we hypothesized that these changes in gene and protein expression would have functional consequences for the immature epithelial barrier. RNA-seq analysis demonstrated broad up-regulation of transcription in genes involved in the formation of the adherens junction and other cell-cell interactions in HIOs after microinjection with live *E. coli* that was inhibited in the presence of NF-κB inhibitor SC-514 (*Figure 8A*). We utilized a modified FITC-dextran permeability assay (*Leslie et al., 2015*) and real-time imaging of live HIO cultures to measure epithelial barrier function in HIOs microinjected with PBS, live *E. coli*, or live *E. coli* +SC-514 at 24 hr after microinjection (*Figure 8B*). While HIOs microinjected with PBS or *E. coli* retained 94.1 0.3% of the FITC-dextran fluorescence over the 20-hr assay period, *E. coli* microinjected HIOs cultured in the presence of SC-514 retained only 45.5 ± 26.3% of the fluorescent signal (p=0.02; *Figure 8B*). We also measured the rate of bacterial translocation across the HIO epithelium, which resulted in contaminated culture media (*Figure 8C*). HIOs microinjected with *E. coli* and treated with SC-514 were compared to *E. coli* microinjected HIOs treated with vehicle (DMSO controls) and PBS microinjected controls over 7 days in culture. HIOs associated with *E. coli* +SC-514 exhibited a rapid onset of bacterial translocation by days 2–3, with bacterial translocation detected in 96% of SC-514-treated HIOs by day 7 compared to 23% of HIOs microinjected with *E. coli* and cultured in DMSO (P=<2 × 10$^{-16}$; *Figure 8C*). Therefore, blocking NF-κB signaling inhibited epithelial barrier maturation resulting in increased bacterial translocation during *E. coli* association with the immature epithelium.

## Bacterial colonization promotes resilience of the epithelial barrier during cytokine challenge

Finally, we assayed epithelial barrier function under circumstances recapitulating physiologic inflammation. TNFα and IFNγ are key cytokines mediating innate and adaptive immune cell activity in the gut (*Turner, 2009*) during bacterial infection (*Rhee et al., 2005*; *Emami et al., 2012*) and in necrotizing enterocolitis (*Tan et al., 1993*; *Ford et al., 1996*, *1997*; *Halpern et al., 2003*; *Upperman et al., 2005*). The combination of TNFα and IFNγ has been previously demonstrated to induce barrier permeability in a dose-dependent manner in Transwell epithelial cultures (*Wang et al., 2005*; *Wang et al., 2006*). Thus, HIOs were microinjected with PBS or live *E. coli* and cultured for 24 hr and were subsequently microinjected with FITC-dextran and treated with PBS or a cocktail of TNFα

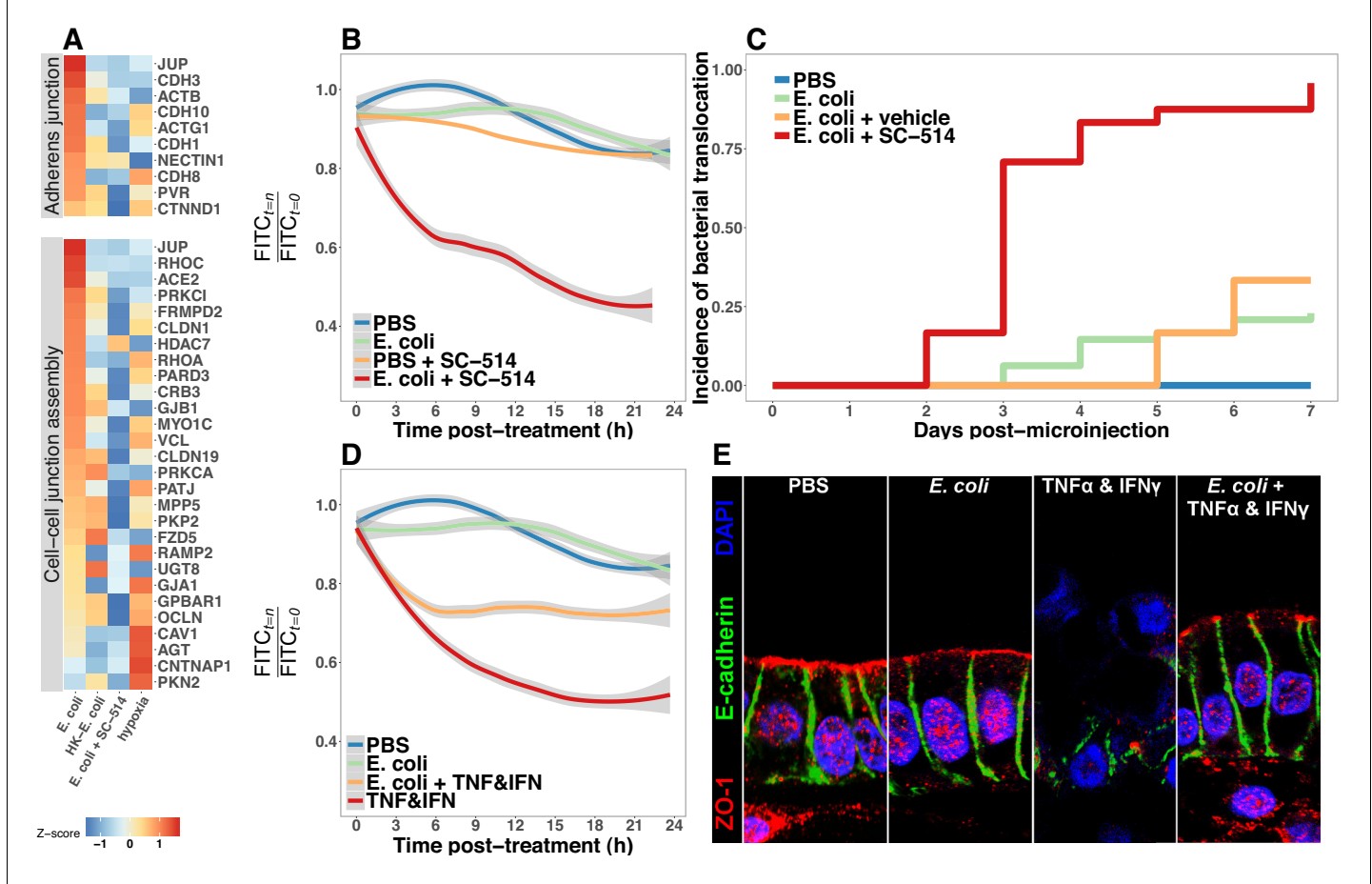

**Figure 8.** Bacterial colonization promotes resilience of the epithelial barrier via NF-κB. (A) Heatmap of RNA-seq data indicating the relative expression of genes associated with the Adherens junction or Cell-cell junction assembly based on annotation in the REACTOME database. Results represent the mean of $N = 4$–5 biological replicates per treatment condition, with each replicate consisting of 5–6 pooled and identically treated HIOs. (B) Relative fluoresscence intensity over time in HIOs microinjected with 4 kDa FITC-dextran and imaged at 10 min intervals. HIOs were pretreated by microinjection with $10^4$ CFU *E. coli* in PBS or PBS alone and cultured for 24 hr prior to treatment with media containing 10 $\mu$M SC-514 or PBS alone and the injection of 2 mg/ml FITC-dextran (4 kDa) at the start of imaging. Line represents the best fit to the mean fluorescent intensity values in each condition with the grey region indicating S.E. for the fit line. $N = 7$–9 HIOs per group. (C) Rate of bacterial translocation over time in HIOs treated as indicated in the figure legend as detected by daily collection of external HIO media and enrichment in bacterial growth broth. $N = 24$ (*E. coli* + SC-514), $N = 48$ (*E. coli*), and $N = 12$ (PBS and *E. coli* + vehicle). (D) Relative fluorescence intensity over time in HIOs microinjected with FITC-dextran and imaged at 10 min intervals. HIOs were pretreated by microinjection with $10^4$ CFU *E. coli* in PBS or PBS alone and cultured for 24 hr prior to treatment with media containing 500 ng/ml TNF-α and 500 ng/ml IFN-γ or PBS alone and the injection of 2 mg/ml FITC-dextran (4 kDa) at the start of imaging. Line represents the best fit to the mean fluorescent intensity values in each condition with the grey region indicating S.E. for the fit line. $N = 8$–9 HIOs per group. (E) Representative confocal micrographs of HIOs treated as indicated in D. Fluorescent immunostaining pseudocoloring applied as indicated in the figure legend. 60X optical magnification with 2X digital zoom. SC-514, small molecule inhibitor of NF-κB ; HK, heat-inactivated; TNF, tumor necrosis factor-α; IFN, interferon-γ.

DOI: https://doi.org/10.7554/eLife.29132.020

The following figure supplement is available for figure 8:

**Figure supplement 1.** Representative confocal micrographs of HIOs treated as indicated.

DOI: https://doi.org/10.7554/eLife.29132.021

and IFNγ added to the external media to expose the basolateral epithelium (*Figure 8D*). Loss of FITC-dextran fluorescence was observed using live-imaging and indicated that treatment with TNFα and IFNγ alone resulted in a rapid and sustained decrease in luminal fluorescence relative to PBS or *E. coli* injected HIOs (p=5 × 10$^{-4}$, *Figure 8D*). However, HIOs associated with *E. coli* prior to addition of the TNFα and IFNγ cocktail retained significantly more fluorescent signal relative to treatment with TNFα and IFNγ alone (p=0.042, *Figure 8D*). We examined expression and distribution of the

tight junction protein ZO-1, and the basal-lateral protein E-cadherin (ECAD) in histological sections taken from PBS and *E. coli*-associated HIOs subjected to TNFα and IFNγ treatment (*Figure 8E*). Compared to controls, the epithelial layer is highly disorganized in HIOs treated with TNFα and IFNγ, with cytoplasmic ZO-1 staining and disorganized ECAD. By contrast, HIOs associated with *E. coli* prior to TNFα and IFNγ treatment retain and organized columnar epithelium with robust apical ZO-1 and properly localized ECAD staining (*Figure 8E*). Similarly, proper localization of additional markers of epithelial barrier integrity occludin (OCLN) and acetylated-tubulin are retained in HIOs associated with *E. coli* during TNFα and IFNγ treatment relative to HIOs treated with TNFα and IFNγ alone (*Figure 8—figure supplement 1*). These results suggest that colonization of the immature epithelium with *E. coli* results in an epithelium that is more robust to challenge by potentially damaging inflammatory cytokines.

## Discussion

The work presented here demonstrates that HIOs represent a robust model system to study the initial interactions between the gastrointestinal epithelium and colonizing microbes that occurs in the immediate postnatal period. Microorganisms introduced into the digestive tract at birth establish an intimate and mutualistic relationship with the host over time (*Costello et al., 2012*; *Palmer et al., 2007*; *Koenig et al., 2011*; *Bäckhed et al., 2015*; *Wopereis et al., 2014*). However, the expansion of bacterial populations in the gut also presents a major challenge to intestinal homeostasis through the exposure to potentially inflammatory MAMPs (*Tanner et al., 2015*; *Renz et al., 2011*), consumption of tissue oxygen (*Glover et al., 2016*; *Espey, 2013*; *Albenberg et al., 2014*), digestion of the mucus barrier (*Marcobal et al., 2013*; *Desai et al., 2016*), and competition for metabolic substrates (*Rivera-Chávez et al., 2016*; *Kaiko et al., 2016*). The mature intestinal epithelium serves as a crucial barrier to microbes that inhabit the lumen and mucosal surfaces (*Artis, 2008*; *Turner, 2009*; *Desai et al., 2016*; *Kelly et al., 2015*; *Cornick et al., 2015*; *Peterson and Artis, 2014*; *Hackam et al., 2013*; *Turner, 2009*). The specific function of the epithelium in adapting to initial microbial colonization, independent of innate and adaptive immune systems, remains unclear due to the lack of appropriate model systems that recapitulate host-microbe mutualism. Clarifying the role of the epithelium in colonization of the digestive tract by microorganisms is essential to understanding the molecular basis of the stable host-microbe mutualism in the mature intestine.

To examine the establishment of host-microbe mutualism, we chose to examine the interaction between the immature epithelium of HIOs and a non-pathogenic strain of *E. coli*. Enterobacteriaceae, including *E. coli*, are abundant in the newborn gut (*Palmer et al., 2007*; *Koenig et al., 2011*; *Bäckhed et al., 2015*; *DIABIMMUNE Study Group et al., 2016*). Several large-scale surveys of microbial composition have demonstrated that *E. coli* are among the most prevalent and abundant organisms in stool samples from newborns (*Bäckhed et al., 2015*; *Koenig et al., 2011*) and in meconium (*Gosalbes et al., 2013*). Non-pathogenic *E. coli* strains may represent ideal model organisms for examining the impact of bacterial colonization of the immature epithelium due to their prevalence in the neonatal population and relevance to natural colonization, extensive characterization, and ease of laboratory manipulation. Microinjection of non-pathogenic *E. coli* into the lumen of HIOs resulted in stable, long-term co-cultures (*Figure 1*). *E. coli* grows rapidly within the HIO lumen (*Figure 1*), reaching densities roughly comparable to populations found in the human small intestine (*Donaldson et al., 2016*) within 24 hr. Furthermore, the HIO is able to sustain this internal microbial population for several days while retaining the integrity of the epithelial barrier (*Figure 1*). Implicit is this observation is the conclusion that immature epithelium, along with a loosely structured mesenchymal layer, is intrinsically capable of adapting to the challenges imposed by colonization with non-pathogenic gut bacteria.

To more closely examine these epithelial adaptations of microbial colonization, we performed transcriptional analysis of this response. HIOs colonized by *E. coli* exhibit widespread transcriptional activation of innate bacterial recognition pathways, including TLR signaling cascades and downstream mediators such as NF-κB (*Figure 2*). The cellular composition of the HIO epithelium is refined following *E. coli* colonization, with a rapid but transient increase in epithelial proliferation preceding a general reduction in the number of immature epithelial progenitor cells and the emergence of mature enterocytes expressing brush border digestive enzymes (*Figure 3*). Together, these results

suggest that bacterial stimuli exert a broad influence on the molecular and cellular composition of the immature epithelium.

Indirect stimuli resulting from microbial activity can also shape epithelial function (*Buffie and Pamer, 2013*), and the transcriptome of *E. coli*-colonized HIOs reflects a cellular response to reduced oxygen availability (*Figure 2*). Reduction of luminal $O_2$ concentration occurs in the neonatal gut (*Gruette et al., 1965*; *Fisher et al., 2013*; *Zheng et al., 2015*), possibly as a result of the consumption of dissolved $O_2$ by the anaerobic and facultative anaerobic bacteria that predominate in the intestinal microbiome in early life (*Espey, 2013*; *Fanaro et al., 2003*; *Favier et al., 2002*; *Palmer et al., 2007*), and the mature intestinal epithelium is hypoxic relative to the underlying lamina propria due to the close proximity to the anaerobic luminal contents (*Glover et al., 2016*; *Kelly et al., 2015*; *Zheng et al., 2015*). We measured luminal oxygen content and epithelial hypoxia in HIOs microinjected with live *E. coli*, finding that luminal oxygen concentration is reduced more than 10-fold relative to the surrounding media. This state of relative hypoxia extends into the epithelium itself and is correlated with increased microbial density (*Figure 4*). Thus, although HIOs lack the network of capillaries that play an essential role in tissue oxygen supply in the intestine, *E. coli*-colonized HIOs recapitulate in vitro the oxygen gradient present at the epithelial interface.

Colonization of the HIO by *E. coli* therefore comprises two broad stimuli: immediate exposure to contact-mediated signals such as MAMPs, and the onset of limiting luminal oxygen and epithelial hypoxia. Although the potential significance of exposure to microbial products in the context of tissue hypoxia is widely recognized in the setting of necrotizing enterocolitis (*Tanner et al., 2015*; *Afrazi et al., 2014*; *Hackam et al., 2013*; *Neu and Walker, 2011*; *Upperman et al., 2005*; *Nanthakumar et al., 2011*), this two factor signaling paradigm has not been well studied as a component of normal intestinal colonization and development. Using the HIO model system, it was possible to design experiments which separately examine the relative impact of microbial contact-mediated signals from microbe-associated hypoxic signals (*Figure 5* and *Figure 5—figure supplement 2*). This approach reveals that the full transcriptional response generated by the HIO following *E. coli* colonization is the product of both contact-dependent and hypoxia-dependent signals, with heat-inactivated *E. coli* or hypoxia alone recapitulating distinct subsets of the changes in gene expression observed in HIOs colonized with live *E. coli* (*Figure 5*). Future studies may examine the role of additional hypoxia-independent live microbe-associated stimuli, such as metabolic products (*Kaiko et al., 2016*) and viability-associated MAMPs (*Sander et al., 2011*), in mediating the epithelial response to initial bacterial colonization.

NF-κB signaling has been implicated in the downstream response to both microbial contact-mediated signals (*Zhang and Ghosh, 2001*; *Xiao and Ghosh, 2005*; *Kawai and Akira, 2007*) and tissue hypoxia (*Koong et al., 1994*; *Rius et al., 2008*; *Arias-Loste et al., 2015*; *Oliver et al., 2009*; *Zeitouni et al., 2016*; *Colgan et al., 2013*; *Grenz et al., 2012*). Pharmacologic inhibition of NF-κB resulted in the suppression of both microbial contact- and hypoxia-associated gene expression in HIOs, inhibiting both contact-mediated epithelial barrier defense pathways and hypoxia-associated immune activation (*Figure 5*). NF-κB appears to play a key role in integrating the complex stimuli resulting from exposure to microbial products and the onset of localized hypoxia in the immature intestinal epithelium during bacterial colonization.

The molecular and cellular maturation of the intestine that occurs during infancy ultimately results in enhanced functional capacity (*Lebenthal and Lebenthal, 1999*; *Sanderson and Walker, 2000*; *Neu, 2007*). Bacterial colonization is associated with enhanced epithelial barrier function in gnotobiotic animals, including changes in the production of antimicrobial peptides and mucus (*Vaishnava et al., 2008*; *Cash et al., 2006*; *Goto et al., 2014*; *García-Lafuente et al., 2001*; *Malago, 2015*; *Ménard et al., 2008*). Defensins produced in the intestinal epithelium are critical mediators of the density and composition of microbial populations in the gut and protect the epithelium from microbial invasion (*Kisich et al., 2001*; *Ostaff et al., 2013*; *Cullen et al., 2015*; *Salzman et al., 2003*; *Salzman et al., 2010*). Production of BD-2 is dramatically increased in HIOs immediately following *E. coli* colonization (*Figure 2*, *Figure 5—figure supplement 2* and *Figure 6*), reaching concentrations that are sufficient to limit overgrowth of *E. coli* (*Figure 6* and *Figure 6—figure supplement 2*) without completely precluding potentially beneficial bacterial colonization (*Figure 1*). Secreted and cell-surface associated mucins form a physical barrier to microbes in the gut, act as local reservoirs of antimicrobial peptide, and serve as substrates for the growth of beneficial microorganisms (*Desai et al., 2016*; *Johansson and Hansson, 2016*; *Cornick et al., 2015*;

*Hansson, 2012*; *Li et al., 2015*; *Dupont et al., 2014*; *Bergstrom and Xia, 2013*). The immature HIO epithelium produces a robust mucus layer consisting of both neutral and acidic oligosaccharides with terminal carbohydrate modifications following colonization with *E. coli* (*Figure 7*). Importantly, hypoxia alone does not result in the production of mucus while the introduction of heat-inactivated *E. coli* induces mucus secretion at the apical epithelium (*Figure 7*), suggesting that microbial contact is the major stimulus eliciting mucus secretion in HIOs.

Epithelial barrier permeability is an important parameter of intestinal function reflecting the degree of selectivity in the transfer of nutrients across the epithelial layer and the exclusion of bacteria and other potentially harmful materials (*Bischoff et al., 2014*). Increases in epithelial barrier permeability occur in the setting of inflammation (*Ahmad et al., 2017*; *Michielan and D'Incà, 2015*) and infectious disease (*Shawki and McCole, 2017*). Colonization of HIOs with *E. coli* results in increased transcription of genes associated with the formation of the adherens junction and other cell-cell interactions in the epithelium (*Figure 8*). However, inhibition of NF-κB signaling dramatically increases both epithelial barrier permeability and the rate of bacterial translocation (*Figure 8*), suggesting that NF-κB signaling is critical to maintaining epithelial barrier integrity following colonization. Expression of genes involved in the formation of the cell junction and the production of antimicrobial defensins and mucus are NF-κB dependent (*Figures 6–8*, *Figure 7—figure supplement 1*, *Tsutsumi-Ishii and Nagaoka, 2002*; *Ahn et al., 2005*). The inability to mount an effective innate defense response in the presence of NF-κB inhibition results in the failure of the HIO epithelial barrier and the loss of co-culture stability (*Figure 8*). This result underscores the critical role of NF-κB signaling in the formation of a stable host-microbe mutualism at the immature epithelial interface.

Dysregulated production of pro-inflammatory cytokines contributes to the loss of epithelial barrier integrity in NEC (*Tanner et al., 2015*; *Hackam et al., 2013*; *Neu and Walker, 2011*; *Nanthakumar et al., 2011*; *Halpern et al., 2003*; *Ford et al., 1997*, *1996*; *Tan et al., 1993*); this is recapitulated in HIOs, as exposure to pro-inflammatory cytokines results in the rapid loss of epithelial barrier integrity and the dissolution of epithelial tight junctions (*Figure 8*). Probiotics may promote epithelial barrier integrity in NEC (*Robinson, 2014*; *Alfaleh et al., 2011*; *Underwood et al., 2014*; *Khailova et al., 2009*) and HIOs colonized by *E. coli* exhibit enhanced epithelial barrier resilience (*Figure 8*). Functional maturation resulting from colonization of the immature intestinal epithelium may therefore play an essential role in promoting the resolution of physiologic inflammation.

While great progress has been made in characterizing the composition of the gut microbiota in health and disease (*Shreiner et al., 2015*; *Costello et al., 2012*), this approach has a limited ability to discern the contributions of individual bacteria to the establishment of host-microbe symbiosis. Our work establishes an approach that recapitulates host-microbe mutualism in the immature human intestine in an experimentally tractable in vitro model system. Application of this approach may facilitate the development of mechanistic models of host-microbe interactions in human tissue in health and disease. For example, one of the major limitations in our understanding of NEC has been the lack of an appropriate model system to study colonization of the immature intestine (*Neu and Walker, 2011*; *Balimane and Chong, 2005*; *Tanner et al., 2015*; *Nguyen et al., 2015*). Our results suggest that colonization of the HIO with a non-pathogenic gut bacteria results in functional maturation of the epithelial barrier. Future work which examines the effects of organisms associated with the premature gut (*Morrow et al., 2013*; *Greenwood et al., 2014*; *Ward et al., 2016*) on the molecular, cellular, and functional maturation of the immature epithelium may be instrumental in elucidating mechanisms of microbiota-associated disease pathogenesis in the immature intestine.

# Materials and methods

## HIO culture

Human ES cell line H9 (NIH registry #0062, RRID:CVCL_9773) was obtained from the WiCell Research Institute. H9 cells were authenticated using Short Tandem Repeat (STR) DNA profiling (*Matsuo et al., 1999*) at the University of Michigan DNA Sequencing Core and exhibited an STR profile identical to the STR characteristics published by (*Josephson et al., 2006*). The H9 cell line was negative for *Mycoplasma* contamination. Stem cells were maintained on Matrigel (BD Biosciences, San Jose, CA) in mTeSR1 medium (STEMCELL Technologies, Vancouver, Canada). hESCs were

passaged and differentiated into human intestinal organoid tissue as previously described (*Spence et al., 2011*; *McCracken et al., 2011*). HIOs were maintained in media containing EGF, Noggin, and R-spondin (ENR media, see [*McCracken et al., 2011*]) in 50 µl Matrigel (8 mg/ml) without antibiotics prior to microinjection experiments. For hypoxic culture experiments, HIOs were transferred to a hydrated and sealed Modular Incubator Chamber (MIC-101, Billups-Rothenburg, Inc. Del Mar CA) filled with 1% $O_2$, 5% $CO_2$, and balance $N_2$ and maintained at 37 for 24 hr.

## HIO transplantation and tissue derived enteroid culture

HIO transplantations: This study was performed in strict accordance with the recommendations in the Guide for the Care and Use of Laboratory Animals of the National Institutes of Health. All animal experiments were approved by the University of Michigan Institutional Animal Care and Use Committee (IACUC; protocol # PRO00006609). HIO transplants into the kidney capsule were performed as previously described (*Finkbeiner et al., 2015*; *Dye et al., 2016*) Briefly, mice were anesthetized using 2% isofluorane. The left flank was sterilized using Chlorhexidine and isopropyl alcohol, and an incision was made to expose the kidney. HIOs were manually placed in a subcapsular pocket of the kidney of male 7- to 10-week-old NOD-SCID IL2Rgnull (NSG) mice using forceps. An intraperitoneal flush of Zosyn (100 mg/kg; Pfizer Inc.) was administered prior to closure in two layers. The mice were sacrificed and transplant retrieved after 10 weeks. Human Tissue: Normal, de-identified human fetal intestinal tissue was obtained from the University of Washington Laboratory of Developmental Biology. Normal, de-identified human adult intestinal tissue was obtained from deceased organ donors through the Gift of Life, Michigan. All human tissues used in this work were obtained from non-living donors, were de-identified and were conducted with approval from the University of Michigan IRB (protocol # HUM00093465 and HUM00105750). Isolation and culture of HIO epithelium, transplanted HIO epithelium, fetal and adult human duodenal epithelium was carried out as previously described (*Finkbeiner et al., 2015*), and was cultured in a droplet of Matrigel using L-WRN conditioned medium to stimulate epithelial growth, as previously described (*Miyoshi et al., 2012*; *Miyoshi and Stappenbeck, 2013*)

## Bacterial culture

*Escherichia coli* strain ECOR2 (ATCC 35321) was cultured in Luria broth (LB) media or 1.5% LB agar plates at 37 under atmospheric oxygen conditions. Glycerol stock solutions are available upon request. The assembled and annotated genome for the isolate of *Escherichia coli* strain ECOR2 used in these studies is available at https://www.patricbrc.org/view/Genome/562.18521. *E. coli* strain K-12 MG1655 (CGSC #6300) was obtained from the Coli Genetic Stock Center at Yale University (http://cgsc2.biology.yale.edu/) and was used only in the in vitro BD-2 activity experiments. Whole genome sequencing of *E. coli* strain ECOR2 was performed by the University of Michigan Host Microbiome Initiative Laboratory using the Illumina MiSeq platform.

## Microinjection

Microinjections were performed using a protocol modified from *Leslie et al., 2015*. Briefly, HIOs were injected using thin wall glass capillaries (TW100F-4, World Precision Instruments, Sarasota, FL) shaped using a P-30 micropipette puller (Sutter Instruments, Novato, CA). Pulled microcapilaries were mounted on a Xenoworks micropipette holder with analog tubing (BR-MH2 and BR-AT, Sutter Instruments) attached to a 10-ml glass syringe filled with sterile mineral oil (Fisher Scientific, Hampton, NH). Fine control of the micropippette was achieved using a micromanipulator (Narishge International Inc., East Meadow, NY) and microinjection was completed under 1-2X magnification on an SX61 stereo dissecting scope (Olympus, Tokyo, Japan). HIOs suspended in Matrigel (Corning Inc., Corning, NY) were injected with approximately 1 µl solution. A detailed and up-to-date HIO microinjection protocol is available at *Hill, 2017b*; a copy is archived at https://github.com/elifesciences-publications/HIO_microinjection). In bacterial microinjection experiments, the HIO culture media was removed immediately following microinjection and the cultures were rinsed with PBS and treated with ENR media containing penicillin and streptomycin to remove any bacteria introduced to the culture media during the microinjection process. After 1 hr at 37, the HIOs were washed again in PBS and the media was replaced with fresh antibiotic-free ENR.

## Measurement of luminal oxygen

Luminal oxygen content was measured in HIOs using an optically coated implantable microsensor with a tip tapered at <50 μm (IMP-PSt1, PreSens Precision Sensing GmbH) attached to a micro fiber optic oxygen meter (Microx TX3, PreSens Precision Sensing GmbH, Regensburg, Germany). The oxygen probe was calibrated according to the manufacturer's instructions and measurements of the external media and HIO luminal oxygen content were collected by mounting the microsensor on a micromanipulator (Narishge International Inc., East Meadow, NY) and guiding the sensor tip into position using 1-2X magnification on a stereo dissecting scope (Olympus, Tokyo, Japan). All oxygen concentration readings were analyzed using PreSens Oxygen Calculator software (TX3v531, PreSens Precision Sensing GmbH, Regensburg, Germany). For measurement of relative cytoplasmic hypoxia, HIO cultures were treated with 100 μM pimonidazol HCl (Hypoxyprobe, Inc., Burlington, MA) added to the external culture media and incubated at 37% and 5% $CO_2$ for 2 hr prior to fixation in 4% parafomaldehyde. Pimonidazole conjugates were stained in tissue sections using the Hypoxyprobe-1 mouse IgG monoclonal antibody (Hypoxyprobe, Inc., Burlington, MA, RRID:AB_2335667) with appropriate secondary antibody (see antibody dilutions table).

## Immunohistochemistry

Immunostaining was carried out as previously described (*Finkbeiner et al., 2015*). Antibody information and dilutions can be found in *Supplementary file 2*. All images were taken on a Nikon A1 confocal microscope or an Olympus IX71 epifluorescent microscope. CarboFree blocking buffer (SP-5040; Vector Laboratories, Inc. Burlingame, CA) was substituted for dilute donkey serum in PBS in staining for mucins and carbohydrate moieties. EdU treatment and EdU fluorescent labeling using Click-iT chemistry was applied according to the manufacturer's instructions (#C10339 Thermo Fisher, Waltham, MA). *Supplementary file 2* contains a table of all primary and secondary antibodies, blocking conditions, and product ordering information.

## NF-κB inhibition

The NF-κB inhibitor SC-514 (*Kishore et al., 2003*; *Litvak et al., 2009*) (Tocris Cookson, Bristol, UK) was re-suspended in DMSO at a concentration of 25 mM. HIOs were treated with SC-514 suspended in DMSO added to the external ENR culture media at a final concentration of 1 μM. Efficacy of SC-514 was verified by Western blot of lysates from HIOs injected with PBS or live *E. coli* or injected with live *E. coli* in the presence of 1 μM SC-514 added to the external media. HIOs were collected after 24 hr in lysis buffer composed of 300 mM NaCl, 50 mM Tris base, 1 mM EDTA, 10% glycerol, 0.5% NP-40, and 1X Halt Phosphatase Inhibitor Cocktail (Pierce Biotechnology, Rockford, IL). Lysates were separated on a 10% Bis-Tris polyacrylamide gel under reducing conditions (Invitrogen, Carlsbad, CA) and transferred to PVDF using a wet transfer apparatus (Bio-Rad Laboratories, Hercules, CA) overnight at 4. The PVDF membrane was blocked in Odyssey TBS blocking buffer (LI-COR Biosciences, Lincoln, NE). The membrane was submerged in blocking buffer containing primary rabbit monoclonal antibodies against phosphorylated NF-κB p65 (1:200, Cell Signaling Technology #3033S) or total NF-κB p65 (1:400, Cell Signaling Technology #8242S) and incubated at room temperature for 2 hr. All washes were conducted in Tris-buffered saline with 1% Tween-20 (TBST). The secondary goat anti-rabbit IgG IRDye 800CW was diluted 1:15,000 in TBST and exposed to the washed membrane for 1 hr at room temperature. After additional washes, the PVDF membrane was imaged using an Odyssey Scanner (LI-COR Biosciences, Lincoln, NE).

## Bacterial translocation assay

Incidence of bacterial translocation was determined in HIOs plated individually in single wells of 24-well plates and microinjected with *E. coli*. The external culture media was collected and replaced daily. The collected media was diluted 1:10 in LB broth in 96 well plates and cultured at 37 overnight. Optical density (600 nm) was measured in the 96-well LB broth cultures using a VersaMax microplate reader (Molecular Devices, LLC, Sunnyvale, CA). $OD_{600}$> sterile LB broth baseline was considered a positive culture.

## FITC-dextran permeability

For epithelial permeability assays, HIOs were microinjected with 4 kDa FITC-dextran suspended in PBS at a concentration of 2 mg/ml as described previously (*Leslie et al., 2015*) using the microinjection system detailed above. Images were collected at 10 min intervals at 4X magnification on an Olympus IX71 epifluorescent microscope using a Deltavision RT live cell imaging system with Applied Precision softWoRx imaging software (GE Healthcare Bio-Sciences, Marlborough, MA). Cultures were maintained at 37% and 5% $CO_2$ throughout the imaging timecourse. For experiments involving cytokine treatment, recombinant TNF-α (#210-TA-010, R and D Systems) and INF-γ (#AF-300–02, Peprotech) were added to the external culture media at a concentration of 500 ng/ml at the start of the experiment. A detailed and up-to-date HIO microinjection and live imaging protocol is available at (*Hill, 2017b*).

## In vitro antimicrobial activity assay

Recombinant human BD-2 (Abcam, Cambridge, MA) was reconstituted in sterile LB broth and diluted to 0.1–1 µg/ml. *E. coli* cultures were diluted 1:1000 in sterile LB containing 0–1 µg/ml BD-2 and transferred to a 96-well microplate. A VersaMax microplate reader (Molecular Devices, LLC, Sunnyvale, CA) was used to measure $OD_{600}$ at 10 min intervals in microplates maintained at 37°C with regular shaking over a 18 hr timecourse. For stationary phase antimicrobial assays, overnight cultures of *E. coli* str. ECOR2 were diluted in PBS containing 1 µg/ml BD-2 or heat-inactivated BD-2 (heated at 120 for > 60 min) and placed in a 37 bacterial incubator for 5 hr. Cultures were then spread on LB agar plates and cultured overnight. The number of CFU was counted manually.

## ELISA assays

Secreted cytokine, antimicrobial peptide, and growth factor concentrations were determined by ELISA (Duosets, R and D Systems, Minneapolis, MN) using the manufacturer's recommended procedures at the Immunological Monitoring Core of the University of Michigan Cancer Center.

## RNA sequencing and analysis

RNA was isolated using the mirVana RNA isolation kit and following the 'Total RNA' isolation protocol (Thermo-Fisher Scientific, Waltham MA). RNA library preparation and RNA-sequencing (single-end, 50 bp read length) were performed by the University of Michigan DNA Sequencing Core using the Illumina Hi-Seq 2500 platform. All sequences were deposited in the EMBL-EBI ArrayExpress database (RRID:SCR_004473) using Annotare 2.0 and are cataloged under the accession number E-MTAB-5801. Transcriptional quantitation analysis was conducted using 64-bit Debian Linux stable version 7.10 ('Wheezy'). Pseudoalignment of RNA-seq data was computed using kallisto v0.43.0 (*Bray et al., 2016*) and differential expression of pseudoaligned sequences was calculated using the R package DEseq2 (*Love et al., 2014*) (RRID:SCR_000154).

## Statistical analysis

Unless otherwise indicated in the figure legends, differences between experimental groups or conditions were evaluated using an unpaired Student's *t*-test. A p-value <0.05 was considered to represent a statistically significant result. All statistical analyses were conducted using R version 3.4.1 (2017-06-30) (*Core Team, 2017*) and plots were generated using the R package ggplot2 (*Wickham, 2009*) with the ggstance expansion pack (*Henry et al., 2016*). The multiple testing-adjusted FDR was calculated using the DESeq2 implementation of the Wald test (*Love et al., 2014*). Gene pathway over-representation tests and Gene Set Enrichment Analysis (*Subramanian et al., 2005*) were implemented using the R packages clusterProfiler (*Yu et al., 2012*) and ReactomePA (*Yu and He, 2016*). Analyses conducted in R were collated using Emacs v25.2 (*Stallman, 1981*) with Org-mode v8.3.5 and the paper was written in LaTeX using Emacs. Complete analysis scripts are available on the *Hill, 2017a* GitHub repository (copy archived at https://github.com/elifesciences-publications/Hill_HIO_Colonization_2017).

## Acknowledgements

The authors would like to thank Joel Whitfield of the Immunological Monitoring Core of the University of Michigan Cancer Center, Robert Lyons and Tricia Tamsen of the University of Michigan DNA Sequencing Core, Chris Edwards of the University of Michigan Molecular Imaging Laboratory, and April Cockburn and Micah Kiedan of the University of Michigan Host Microbiome Initiative Laboratory for providing invaluable technical assistance. JRS is supported by the Intestinal Stem Cell Consortium (U01DK103141), a collaborative research project funded by the National Institute of Diabetes and Digestive and Kidney Diseases (NIDDK) and the National Institute of Allergy and Infectious Diseases (NIAID). JRS and VBY are supported by the NIAID Novel Alternative Model Systems for Enteric Diseases (NAMSED) consortium (U19AI116482). DRH is supported the Mechanisms of Microbial Pathogenesis training grant from the National Institute of Allergy and Infectious Disease (NIAID, T32AI007528) and the National Center for Advancing Translational Sciences (UL1TR000433).

## Additional information

### Funding

| Funder | Grant reference number | Author |
|---|---|---|
| National Institute of Allergy and Infectious Diseases | U19AI116482 | Vincent B Young |
| National Institutes of Health | U01DK103141 | Jason R Spence |
| National Institute of Allergy and Infectious Diseases | T32AI007528 | David R Hill |
| National Center for Advancing Translational Sciences | UL1TR000433 | David R Hill |

The funders had no role in study design, data collection and interpretation, or the decision to submit the work for publication.

### Author contributions

David R Hill, Conceptualization, Data curation, Software, Formal analysis, Investigation, Visualization, Methodology, Writing—original draft, Writing—review and editing; Sha Huang, Resources, Investigation, Methodology; Melinda S Nagy, Visualization, Methodology; Veda K Yadagiri, Formal analysis, Methodology; Courtney Fields, Dishari Mukherjee, Brooke Bons, Resources, Methodology; Priya H Dedhia, Alana M Chin, Yu-Hwai Tsai, Methodology; Shrikar Thodla, Investigation, Methodology; Thomas M Schmidt, Seth Walk, Resources; Vincent B Young, Conceptualization, Resources, Supervision, Funding acquisition, Project administration, Writing—review and editing; Jason R Spence, Conceptualization, Supervision, Funding acquisition, Methodology, Writing—review and editing

### Author ORCIDs

David R Hill http://orcid.org/0000-0002-1626-6079
Alana M Chin http://orcid.org/0000-0002-9567-6825
Vincent B Young http://orcid.org/0000-0003-3687-2364
Jason R Spence http://orcid.org/0000-0001-7869-3992

### Ethics

Human subjects: Normal, de-identified human fetal intestinal tissue was obtained from the University of Washington Laboratory of Developmental Biology. Normal, de-identified human adult intestinal issue was obtained from deceased organ donors through the Gift of Life, Michigan. All human tissue used in this work was de-identified and was conducted with approval from the University of Michigan IRB (protocol # HUM00093465 and HUM00105750).

Animal experimentation: This study was performed in strict accordance with the recommendations in the Guide for the Care and Use of Laboratory Animals of the National Institutes of Health. All animal experiments were approved by the University of Michigan Institutional Animal Care and Use Committee (IACUC; protocol # PRO00006609).

Decision letter and Author response
Decision letter https://doi.org/10.7554/eLife.29132.030
Author response https://doi.org/10.7554/eLife.29132.031

## Additional files

### Supplementary files

• Supplementary file 1 List of differentially expressed genes and the Gene Set (I-IV) assignments used in the analyses presented in *Figure 5*.
DOI: https://doi.org/10.7554/eLife.29132.022

• Supplementary file 2 Immunostaining conditions for all immunohistochemistry data presented in this manuscript. NDS, Normal donkey serum; PBS, phosphate buffered saline.
DOI: https://doi.org/10.7554/eLife.29132.023

• Transparent reporting form
DOI: https://doi.org/10.7554/eLife.29132.024

### Major datasets

The following dataset was generated:

| Author(s) | Year | Dataset title | Dataset URL | Database, license, and accessibility information |
|---|---|---|---|---|
| Hill DR | 2017 | RNA-seq of human intestinal organoids colonized with E. coli and other immature intestinal tissues | https://www.ebi.ac.uk/arrayexpress/experiments/E-MTAB-5801/ | Publicly available at the (accession no: E-MTAB-5801) |

The following previously published dataset was used:

| Author(s) | Year | Dataset title | Dataset URL | Database, license, and accessibility information |
|---|---|---|---|---|
| Finkbeiner S, Spence JR | 2014 | Transcriptional Profiling of human pluripotent stem cells and and derived tissues | https://www.ebi.ac.uk/arrayexpress/experiments/E-MTAB-3158/ | Publicly available at the (accession no: EMDataBankE-MTAB-3158) |

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
