## [Decision Letter]

Thank you for submitting your article "Bacterial colonization stimulates a complex physiological response in the immature human intestinal epithelium" for consideration by *eLife*. Your article has been reviewed by three peer reviewers, and the evaluation has been overseen by a Reviewing Editor and Wendy Garrett as the Senior Editor. One of the reviewers, Emma Slack, has agreed to share her identity.

The reviewers have discussed the reviews with one another and the Reviewing Editor has drafted this decision to help you prepare a revised submission. There was consensus among the reviewers that your manuscript would be most appropriate, if adequately revised, for our Tools and Resources section.

Summary:

Human intestinal organoids (HIOs) were used to investigate host microbial interactions at the epithelial surface. Using live or dead *Escherichia coli* they report contact and hypoxia driven responses with antimicrobial peptide production, maturation of the mucus layer and improved paracellular barrier function. The paper represents a technical development of the HIO system, with findings that are aligned with interpretations from previous culture systems.

Essential revisions:

*General comments on presentation in the paper*

Especially for a paper where the main point is development of an experimental system, the technical validation and reporting of the methods and results in the paper must be completely solid. The reviewers noted missing information on the number of biological and technical repeats, concentrations of stimuli, number of injected bacteria from the figure legends. The Materials and methods section was considered incomplete to allow a researcher from a related field to exactly reproduce every experiment shown – concentrations, solvents and timings – e.g. Figure 7, how much TNF, how much IFNgamma? It was unclear exactly how the false discovery rate was assessed – which algorithm was used? What sample numbers were used in the calculations and how reproducible were the experiments? Specific information about the *E. coli* strain should be given, and whether it is motile.

The model depends on selective injection of live bacteria into HIOs and evidence is provided in Figure 1 about this. The experimental protocol needs to be far more explicit in the main body of the text: in the Materials and methods under 'HIO culture' it is stated 'HIOs were maintained in ENR media without antibiotics prior to microinjection experiments'; then under 'Microinjection', '…cultures were rinsed with PBS and treated with ENR media containing penicillin and streptomycin…'. Controls that assess the potential influence of carry-over of antibiotics of the readouts reported in Figure 1 are required.

Further issues on Figure 1

A) Figure 1 appears to be compromised by fixation and/or freezing artifacts.

B) In Figure 1 the y axis is labelled 24h δ CFU. Given that the scale becomes fractional, it seems likely that ratios rather than differences are being used.

C) What dose(s) of injections were used for 1D?

D) As a main panel of Figure 1, showing two agar plates does not add much, especially when the legend does not give details of exactly when are how they were generated. Further, the final sentence of the third paragraph in subsection “HIOs can be stably associated with commensal *E. coli*” and the results quoted in the following paragraph after day 3 are contradictory. Controls are absolutely necessary to ensure that these findings are not simply due to antibiotic carry-over.

The transcriptomic approach does not capture all basic information about the system. Does the *E. coli* injection alter the size, morphology and longevity of the HIOs? Does it alter the rate of epithelial cell turnover (which could be easily tested by BrdU labeling, and would be predicted from e.g. Proc Natl Acad Sci U S A. 2011 Mar 15;108 Suppl 1:4570-7)? Does it alter the pattern of epithelial cell maturation, i.e. the fraction of mature paneth cells, goblet cells, enteroendocrine cells etc?

It is accepted that enteroids are 3 dimensional structures, and the study has been productive in the evaluation of stem cell dynamics and other tissue level events. However, organoids are not vascularized with a capillary network that generate O2 gradients in vivo, thus organism level physiological events such as hypoxia are difficult to address and these caveats must be included in the discussion

Using the pathway analysis from the GO and REACTOME databases limits the insight that the reader is given into the changes in gene expression. For example, in Figure 2 'muscle cell differentiation' is shown as a pathway. It would be possible to report detailed network analyses at least in the supporting material so that the contribution of different transcripts to the overall analysis is clear. This would allow considerable refinement of the time-dependent transcriptional response descriptions in the text.

In Figure 3 it is difficult to appreciate whether the cadherin stain has worked in the PBS panel. How reproducible are these results? What happens with a hypoxic control?

The data analysis shown in Figure 4 was complex and challenging for the reviewers to appreciate in its current form. The subset organisation must be clear and specific.

A) Why is there no PBS control with the NFkB inhibitor? Many cell survival/cell death pathways converge on NFkB signalling and the inhibitor may exert effects already at baseline which would then fall into gene set 1 and gene set 2? From the Venn-diagrams shown in panel A, it appears that genes in set 1 and 2 are mutually exclusive, which would make the lacking control less problematic, but apparently the hypoxia-responsive genes are highly enriched in both (of course these may be different genes in the same pathway? But this would be slightly surprising if a statistical exclusion had been made).

B) For the discussion of the analysis, it appears that an assumption is made that the effect of live *E. coli* is identical to dead *E. coli* combined with hypoxia. There is a large body of literature on "vital PAMPs" (e.g. Nature. 2011 May 22;474(7351):385-9), which might suggest there is more to it. Nevertheless, there appears to be an excellent correlation between genes regulated by live and dead *E. coli* in the plots in Figure 4. If the hypothesis is correct, then injecting dead *E. coli* and then immediately transferring the organoids to a hypoxic chamber should produce a gene-expression profile that correlates better that the dead *E. coli* alone. Noticeably the gradient of the correlation between hypoxia/PBS and live *E. coli*/PBS seems to be close to zero?

C) Is there some sort of statistical significance cut-off for the genes identified for each set in panel B? The clouds appear to very closely approach a log2-FC of zero, suggesting genes showing very small changes in expression are included? Would it be logical to show the data pre-filtered for p-value? Or show the limits of the region and color-code for p-value?

D) In panel C, is it right that "% genes matched to pathway" is this the percentage of genes from each set (e.g. the full 1940 genes in Set 1) that map to the indicated pathway? Thus 5%, i.e. around 400 genes from set 1 map to "regulation of cytoskeleton organization"? Please clarify. Also, as the plots in B suggest that many genes are included with a very small up- or down-regulation, it would be important to have some handle on not just the significance, but also the average absolute size of the change observed. A second set of graphs, or a supplementary figure with more information would be helpful.

E) In the legend, were pathways with enrichment P-values greater than 0.01 excluded?

In Figure 5, it would be important to include a group microinjected with dead *E. coli*, immediately followed by hypoxia, to conclude that both factors act together to induce b-defensins. For Figure 5, does hBD-2 require a reducing agent for activity as hBD1 does (Nature 469, 419-423)? Typically, these pore-forming AMPs to exert a stronger effect on the rapidly growing bacteria than in the stationary phase, and in fact, your maximum growth rate (i.e. maximum curve gradient) is even higher where the BD-2 is added, suggesting that something in the BD-2 may even permit faster *E. coli* growth in LB. Death over several hours in late stationary phase may be rather due to accumulation of a toxic metabolite. To control for these effects, it will be important to show growth data with heat-inactivation of the BD-2. To focus on killing in the stationary phase a late stationary-phase culture could be treated with differing concentrations of BD-2 over short time-courses (including the inactivated controls), measuring loss of membrane integrity by Sytox-green uptake by flow cytometry or microscopy. As O-antigens can inhibit AMP function, *E. coli* K-12 could be included in these experiments.

In relation to Figure 6, can induction of mucus production and induction of goblet cell differentiation be delineated? The slow appearance of mucin gene upregulation appears more consistent with a differentiation phenotype than simple gene expression?

The interpretation that 'Epithelial barrier integrity is enhanced following bacterial association' (p13) is rather at odds with the data in Figure 7 where the PBS and *E. coli* treated permeability is the same. Is the meaning that NFkappaB signaling is required for the compensatory effects of the barrier in *E. coli*-treated organoids? What is the effect of the inhibitor alone?

---

## [Author Response]

General comments on presentation in the paper

Especially for a paper where the main point is development of an experimental system, the technical validation and reporting of the methods and results in the paper must be completely solid. The reviewers noted missing information on the number of biological and technical repeats, concentrations of stimuli, number of injected bacteria from the figure legends. The Materials and methods section was considered incomplete to allow a researcher from a related field to exactly reproduce every experiment shown – concentrations, solvents and timings – e.g. Figure 7, how much TNF, how much IFNgamma?

Thank you for highlighting our oversight to include more details on the experimental methods and design. All figure legends have been revised to include the number of replicates and relevant details regarding the experimental setup. Given that microinjecting HIOs is not trivial, we had decided to write a methods paper dedicated to describing this technique in painful detail. Thus, we recently submitted a revised version of a manuscript describing the methodology for microinjecting and measuring FITC-dextran permeability in HIOs, which was accepted at the Journal of Visualized Experiments today (see attached letter of acceptance). We have deposited a pre-print version of this manuscript, which is available at https://github.com/hilldr/HIO_microinjection. We have referenced to this website in the Materials and methods with the hopes that interested readers will have access to all of the details necessary to repeat our procedures. Finally, recognizing the Editor's decision to consider this manuscript for the Tools and Resources section, we have devised a schematic illustrating our experimental setup for the HIO colonization experiments (Figure 1—figure supplement 3). We hope that this will help the reader conceptualize the experimental scheme, evaluate the experiments presented, and devise new experiments to follow up on this work.

It was unclear exactly how the false discovery rate was assessed – which algorithm was used? What sample numbers were used in the calculations and how reproducible were the experiments?

The multiple testing-adjusted FDR was calculated using the DESeq2 (Love et al., 2014) implementation of the Wald test, a method for differential analysis of count data using shrinkage estimation for dispersions and fold changes to improve stability and interpretability of estimates. This allows us to incorporate the degree of uncertainty present in the alignment of any one 50 bp RNA-seq read to the annotated genome, so that counts of reads per gene are weighted according to the level of confidence in the alignment. An additional comment has been added to the Materials and methods to clarify this. The complete analysis and custom scripts have been made available at

https://github.com/hilldr/Hill_HIO_Colonization_2017, and we have also noted access to this information in the Materials and methods section.

Specific information about the E. coli strain should be given, and whether it is motile.

Regarding the *E. coli* strain ECOR2 that was used for these studies, we have performed whole geneome sequencing, annotation, and assembly. Details are given in the Materials and methods and this data is available using the PATRIC online bacterial genomics platform. Phylogenetic analysis of the complete genome (Figure 1—figure supplement 2) indicates that ECOR2 is closely related to the well-studied *E. coli* type strain K-12 MG1655 and other nonpathogenic *E. coli* isolates. *E. coli* strain ECOR2 is available from ATCC. We have also indicated in the manuscript that we are willing to share isolates of this organism with interested researchers pending appropriate permissions of ATCC, who owns the strain.

The model depends on selective injection of live bacteria into HIOs and evidence is provided in Figure 1 about this. The experimental protocol needs to be far more explicit in the main body of the text: in the Materials and methods under 'HIO culture' it is stated 'HIOs were maintained in ENR media without antibiotics prior to microinjection experiments'; then under 'Microinjection', '…cultures were rinsed with PBS and treated with ENR media containing penicillin and streptomycin…'. Controls that assess the potential influence of carry-over of antibiotics of the readouts reported in Figure 1 are required.

In the interest of brevity, we originally aimed to write a concise description of the methodology. However, clarity is more important than conciseness in this case, particularly given the Editor's proposal to publish this work under the Tools and Resources section. We have made a concerted effort to improve the level of detail in our reported HIO culture methods. Notably, we have added a supplement to Figure 1 (Figure 1—figure supplement 3) containing a schematic representation of our bacterial microinjection and co-culture scheme and data supporting our methodology. As illustrated in Figure 1—figure supplement 3, by comparing LB vs PBS as a vehicle for microinjection, we determined that the solution used to dilute *E. coli* cultures for microinjections could have a marked effect on the growth of *E. coli* within the HIO. Thus, all *E. coli* microinjections were performed using *E. coli* diluted in PB. Furthermore, we show that while maintaining antibiotics in HIO cultures for the duration of a co-culture experiment does not prevent *E. coli* growth in the HIO lumen, the average *E. coli* density was significantly reduced in HIO cultures in the presence of antibiotics (Figure 1—figure supplement 3). As a result, we decided to perform only a brief 1 h antibiotic incubation immediately following microinjection to limit potential growth in the media outside of the HIO. However, this raised concerns that antibiotic carryover could potentially account for the apparent low rate bacterial translocation in HIOs microinjected with *E. coli.* To address this concern, we tested the bacterial growth inhibition potential of culture media collected from HIOs during the 1 h antibiotic incubation that follows microinjection relative to HIO media samples collected after a PBS wash and replacement with fresh, antibiotic-free media. ENR media containing antibiotics precluded growth of *E. coli* str ECOR2, while antibiotic-free media collected after the culture washout had no inhibitory effect on *E. coli* growth (Figure 1—figure supplement 3). Furthermore, antibiotic-free HIO media was changed daily for the experiments presented in Figure 1 and Figure 8, meaning that any carryover antibiotic activity (undetectable in the experiments presented in Figure 1—figure supplement 3) would be reduced many fold within a day or two. By performing these additional experiments to validate our experimental workflow, we are now more confident that our results reflect the influence of the bacterial interaction with HIO, and are not influenced by antibiotic carryover.

Further issues on Figure 1

A) Figure 1 appears to be compromised by fixation and/or freezing artifacts.

This was a good observation, as the data presented was taken from a frozen section. We have replaced this figure with a high magnification confocal micrograph of an HIO at 24 hours after microinjection with live *E. coli* from a separate experiment, and from a sample that was processed through paraffin embedding and sectioning. We hope the use of this alternative data will allow the reader to see the physical relationship between the HIO and *E. coli* with greater clarity.

B) In Figure 1 the y axis is labelled 24h δ CFU. Given that the scale becomes fractional, it seems likely that ratios rather than differences are being used.

The reviewers are correct in pointing out that the y-axis label in Figure 1 was inaccurate. This has been amended in the revised manuscript. In addition, Figure 1—figure supplement 2 has been provided to clarify the exact relationship between the input CFU at t=0 and the CFU harvested from the lumen at 24 hours post microinjection.

C) What dose(s) of injections were used for 1D?

For Figure 1, the HIOs were microinjected with 10 CFU *E. coli* each. The figure legend has been amended to fix this.

D) As a main panel of Figure 1, showing two agar plates does not add much, especially when the legend does not give details of exactly when are how they were generated. Further, the final sentence of the third paragraph in subsection “HIOs can be stably associated with commensal E. coli” and the results quoted in the following paragraph after day 3 are contradictory. Controls are absolutely necessary to ensure that these findings are not simply due to antibiotic carry-over.

Figure 1 has been removed from the main figure. Our intended purpose was to illustrate the growth of *E. coli* within the HIO lumen and the absence of growth in the external media, findings that are made clear in Figure 1 (now 1E).

The transcriptomic approach does not capture all basic information about the system. Does the E. coli injection alter the size, morphology and longevity of the HIOs? Does it alter the rate of epithelial cell turnover (which could be easily tested by BrdU labeling, and would be predicted from e.g. Proc Natl Acad Sci U S A. 2011 Mar 15;108 Suppl 1:4570-7)? Does it alter the pattern of epithelial cell maturation, i.e. the fraction of mature paneth cells, goblet cells, enteroendocrine cells etc?

While the transcriptional analysis demonstrated a time-dependent change in the molecular content of the cells that comprise the HIO following *E. coli* colonization, bacteria could alter the cellular composition of the HIO tissue as well. As mentioned by the reviewers, previous studies have demonstrated that bacterial colonization promotes epithelial proliferation in model organisms. We conducted an experiment to examine epithelial proliferation and differentiation in HIOs over a timecourse of 96 hours, resulting in a new figure being added to the main text (Figure 3). The number of proliferating epithelial cells was elevated by as much as 3-fold in *E. coli*-colonized HIOs relative to PBS-treated HIOs at 24 h. However, at 48 h post-microinjection, the proportion of proliferating epithelial cells was significantly decreased in *E. coli* colonized HIOs relative to control treated HIOs. This observation was supported by RNA-seq data demonstrating an overall suppression of cell cycle genes in *E. coli* colonized HIOs relative to PBS-injected HIOs at 48 h post-microinjection (Figure 3—figure supplement 1). By 96 h post-microinjection the proportion of EdU+ epithelial cells was nearly identical in *E. coli* and PBS-treated HIOs (Figure 3). These results suggest that *E. coli* colonization is associated with a rapid burst of epithelial proliferation, but that relatively few of the resulting daughter cells are retained subsequently within the epithelium.

We also examined *Sox9* expression in HIOs following microinjection with *E. coli* or PBS alone. *Sox9* is known to mark the progenitor domain in the immature intestine, and its expression was dramatically reduced in *E. coli*-colonized HIOs at 48-96 h after microinjection and was notably distributed in distinct clusters directly adjacent to the underlying mesenchyme. This observation indicates an overall reduction in the number of progenitor cells in the HIO epithelium following *E. coli* colonization and implies that other epithelial types may account for a greater proportion of the HIO epithelium at later time points postcolonization. Although we saw no difference in the proportion of epithelial cells expressing goblet, Paneth, or enteroendocrine cell markers (MUC2, DEFA5, and CHGA, respectively), expression of the enterocyte brush border enzyme dipeptidyl peptidase-4 (DPPIV) was found only in the *E. coli*-colonized HIOs at 48 and 96 h post-microinjection. We interpret these data to mean that /*E. coli*/ colonization induces a transient increase in the rate of epithelial proliferation followed by a subsequent reduction in proliferation, progenitor cells and an increase in the enterocyte-like population.

*It is accepted that enteroids are 3 dimensional structures, and the study has been productive in the evaluation of stem cell dynamics and other tissue level events. However, organoids are not vascularized with a capillary network that generate O2 gradients* in vivo*, thus organism level physiological events such as hypoxia are difficult to address and these caveats must be included in the discussion.*

In the intestine oxygen is supplied to the epithelium via a network of capillaries that extend into the lamina propria underlying the epithelium. Recent work has demonstrated that healthy intestinal epithelium exists in a state of relative hypoxia in comparison to the underlying mucosae due to the impact of the anaerobic luminal environment (Glover, Lee and Colgan, 2016, Kelly et al., 2015, Schmidt and Kao, 2014), although the molecular mecahnisms that allow the epithelium to fluorish under these conditions remain unclear. Stem cell derived human intestinal organoids (or enteroids) lack endothelial tissue and oxygen is apparently supplied to the tissue via diffusion through the media, similar to traditional cell culture systems. However, our work suggests that HIOs may be an excellent model system for studying the hypoxia gradient that exists between the lumen and the intestinal epithelium and underlying lamina propria and the changes in the relative state of oxygenation that occur during bacterial colonization (Figure 4) and under other conditions such as inflammation. We have added comments to the discussion clarifying the distinction between localized epithelial hypoxia and tissue hypoxia that occurs as a result of changes in blood flow, which cannot be modeled in the HIO at present.

Using the pathway analysis from the GO and REACTOME databases limits the insight that the reader is given into the changes in gene expression. For example, in Figure 2 'muscle cell differentiation' is shown as a pathway. It would be possible to report detailed network analyses at least in the supporting material so that the contribution of different transcripts to the overall analysis is clear. This would allow considerable refinement of the time-dependent transcriptional response descriptions in the text.

With the GSEA analysis presented in Figure 2, we aimed to present a broad scale overview of the coordinated, time-dependent transcriptional response to *E. coli* colonization based on GO and REACTOME databases. Gene-level pathway analysis may compliment this approach by illustrating the interactions between genes and suggesting mechanistic relationships. Based on the reviewer's suggestion, we have performed a variety of gene level analysis of previously annotated KEGG pathways using the Pathview software package (R/Bioconductor). This resulted in a pathway diagram demonstrating broad reduction in the expression of cell cycle genes at 48 h post-microinjection in *E. coli* colonized HIOs, consistent with our findings in Figure 3, and has been included as Figure 3—figure supplement 1 in the revised manuscript. We thank the reviewers for this valuable suggestion.

The KEGG, GO and REACTOME databases are valuable resources for understanding the relationships between genes and interpreting large scale transcriptional changes. However, these tools are inherently limited to known pathways and may fail to identify novel interactions between networks of genes. We performed a gene regulatory network inference analysis of our *E. coli* colonization timecourse data using the method described by Simoes and EmmertStreib (*PLOS* 2012). This analysis identified a large network of genes that are dynamically expressed over 24-96 h post-microinjection that was highly enriched for tissue development, metabolism, carbohydrate transport, and glycotransferases. A preliminary figure illustrating this network is provided below. We feel that this analysis is valuable; however, in order to glean meaningful insights from the data we will need to spend significant additional time interrogating it using these methods. We feel that in its current from it is preliminary and may not add meaningful information to our manuscript. Interested readers will find that the genes included in each of the pathways shown in Figure 2 are listed in the supplemental data table and in the analysis materials at https://github.com/hilldr/Hill_HIO_Colonization_2017.

In Figure 3 it is difficult to appreciate whether the cadherin stain has worked in the PBS panel. How reproducible are these results? What happens with a hypoxic control?

We have repeated the experiment in a new cohort of HIOs and included new representative immunostaining for all conditions, including heat-inactivated *E. coli* and hypoxic culture controls (Figure 4). In addition, we tabulated the number of PMDZ+ HIOs in each condition from two combined experiments and have included this data as a sub-panel of Figure 4. In our hands, the PMDZ staining is quite reliable and demonstrates an apparent decrease in epithelial oxygen content in HIOs injected with live *E. coli* or HIOs subjected to 1% O2 for 24 hours relative to HIOs injected with heat-inactivated *E. coli* or PBS. We are pleased with the results of the control staining and appreciate the reviewers for bringing this oversight to our attention.

The data analysis shown in Figure 4 was complex and challenging for the reviewers to appreciate in its current form. The subset organisation must be clear and specific.A) Why is there no PBS control with the NFkB inhibitor? Many cell survival/cell death pathways converge on NFkB signalling and the inhibitor may exert effects already at baseline which would then fall into gene set 1 and gene set 2?

We included control HIOs injected with PBS and treated with SC-514 in our initial experiment and uploaded the relevant RNA-seq data to the public repository at EMBL (E-MTAB-5801), however we did not address this dimension of the experiment in our original manuscript out of concerns that it might over-complicate the presentation of our analysis. However, in response to concerns from the reviewers, we have added supplemental figures, which we hope will clarify the baseline effects of the NF-κB inhibitor SC-514. Figure 5—figure supplement 1 shows the log2-transformed fold-change in gene expression relative to PBS-injected controls for all 7 experimental conditions examined in this set of experiments: live *E. coli* +/- SC-514, heatkilled *E. coli* +/- SC-514, and hypoxic culture +/- SC-514, and PBS + SC-514. This figure shows that the number of genes altered by treatment with SC-514 alone is comparable to the set of genes altered in other experimental conditions, although there are generally more genes that are down-regulated by SC-514 exposure than there are genes that are upregulated by SC-514. We examined over-represented genes sets from the GO, KEGG, and REACTOME databases in genes that were significantly up- or down-regulated by treatment with SC-514 alone (Figure 5—figure supplement 1) in order to better understand the types of processes that might be influenced by SC-514 exposure. We plotted the pathways that were in the top 90 percentile based on statistical significance, indicating a high degree of enrichment in the gene subsets regulated by NF-κB inhibitor SC-514 at baseline. The data indicates that SC-514 may suppress some aspects of transcription and translation and may up-regulate authophagy and translation-associated processes. Notably, SC-514 does not appear to have a strong effect at baseline on the pathways identified in Figure 5 as key NFkB-dependent responses to bacterial contact and/or hypoxia, namely innate and adaptive defense, epithelial barrier integrity, angiogenesis and hypoxia signaling, or intestinal development.

From the Venn-diagrams shown in panel A, it appears that genes in set 1 and 2 are mutually exclusive, which would make the lacking control less problematic, but apparently the hypoxia-responsive genes are highly enriched in both (of course these may be different genes in the same pathway? But this would be slightly surprising if a statistical exclusion had been made).

This is an important oversight on our part, as the original version of Figure 5 implied that there were no overlapping genes between Gene Set I and Gene Set II. This was not our intention, as it is perfectly feasible that some genes may be induced by either bacterial contact OR hypoxia via NF-κB. In Figure 5—figure supplement 1, we plotted a Venn diagram showing the degree of overlap between Gene Set I, Gene Set II, and the set of genes that are significantly down-regulated in PBS-injected HIOs treated with SC-514. In fact, the Venn diagram shown as Figure 5—figure supplement 1 demonstrates that there are 603 genes that are indeed induced by either hypoxia OR bacterial-contact in an NF-κB dependent manner. This analysis demonstrates that the majority of contact- or hypoxia-induced genes that are NF-κB dependent are not significantly down-regulated in PBS-injected HIOs treated with SC-514 only. However, some genes suppressed by SC-514 at baseline may be biologically significant. Therefore, we examined over-represented genes sets from the GO, KEGG, and REACTOME databases in genes that were shared between Gene Set I or II and the set of genes that are significantly down-regulated in PBS-injected HIOs treated with SC-514 (Figure 5—figure supplement 1). This analysis indicates that the biggest effects of SC-514 at baseline among Gene Set I and Gene Set II genes are related to metabolism, redox state, and transcription/translation. SC-514 may also have an effect of suppressing the response to hypoxia at baseline, which is not surprising given the relatively low oxygen conditions of HIO culture even prior to treatment (Figure 4). Thus, while SC-514 does have some potentially interesting effects on transcription in the HIO even at baseline, the effect of SC-514 alone cannot account for the major NF-κB dependent responses induced by contact and/or hypoxia. We appreciate that the reviewers brought this important point to our attention and we feel that this additional analysis strengthens our conclusion in Figure 5 that NF-κB-dependent responses to bacterial contact and/or hypoxia include innate and adaptive defense, epithelial barrier integrity, angiogenesis and hypoxia signaling, or intestinal development.

B) For the discussion of the analysis, it appears that an assumption is made that the effect of live E. coli is identical to dead E. coli combined with hypoxia. There is a large body of literature on "vital PAMPs" (e.g. Nature. 2011 May 22;474(7351):385-9), which might suggest there is more to it. Nevertheless, there appears to be an excellent correlation between genes regulated by live and dead E. coli in the plots in Figure 4. If the hypothesis is correct, then injecting dead E. coli and then immediately transferring the organoids to a hypoxic chamber should produce a gene-expression profile that correlates better that the dead E. coli alone. Noticeably the gradient of the correlation between hypoxia/PBS and live E. coli/PBS seems to be close to zero?

The reviewers have clarified an underlying implication of our analysis of microbial contact and hypoxia presented in Figure 5: contact with microbial products under hypoxic conditions may recapitulate the effects of colonization with metabolically active live bacteria. Although this condition was not included in our original RNA-seq analysis and the limited time alotted for revision prevents us from completing a follow-up transcriptional analysis, we have examined this hypothesis by evaluating the role of microbial contact and microbe-associated hypoxia in colonization-induced changes in AMP, cytokine, and growth factor secretion (Figure 5—figure supplement 2). The results indicate a complex interplay between hypoxia and bacterial contact stimuli that shapes protein expression during bacterial colonization, with some factors induced by either microbial contact or hypoxia alone (IL-6), other cases in which either bacterial contact or hypoxia appears to be the dominant stimuli (BD-2), and a third regulatory paradigm in which the response to live *E. coli* evidently results from the cumulative influence of bacterial contact and hypoxia (BD-1, IL-8, VEGF). Taken together with our transcriptional analysis, this analysis demonstrates that association of immature intestinal epithelium with live *E. coli* results in a complex interplay between microbial contact and microbe-associated hypoxia induced gene expression and protein secretion.

C) Is there some sort of statistical significance cut-off for the genes identified for each set in panel B? The clouds appear to very closely approach a log2-FC of zero, suggesting genes showing very small changes in expression are included? Would it be logical to show the data pre-filtered for p-value? Or show the limits of the region and color-code for p-value?

We apologize that this was not made more clear in the initial submission. We did use p-value as a pre-filter for this analysis and we also used only genes that were up-regulated in a treatment group, relative to control. We have revised the text to ensure that this information on how we filtered the data is made more explicit.

Since we used p-values as a filter, some genes that had very modest fold change differences were included in the analysis, explaining the log2-FC values that approach zero. We hope that the clarification around this text makes the data presented sufficiently clear.

D) In panel C, is it right that "% genes matched to pathway" is this the percentage of genes from each set (e.g. the full 1940 genes in Set 1) that map to the indicated pathway? Thus 5%, i.e. around 400 genes from set 1 map to "regulation of cytoskeleton organization"? Please clarify.

Yes, that is the correct interpretation of the x-axis label. We have revised the label to read "% genes from input set matched to pathway", which we hope will improve clarity for the reader. The plotted value is the proportion of input genes (i.e. the 1,940 genes in Gene Set I) that are included in the given GO or REACTOME pathway set. Therefore if ~4% of genes in Gene Set I map to "regulation of cytoskeleton organization" this would be the equivalent of 0.04 x 1,940 = 97 genes. Note that many genes are assigned to multiple GO and REACTOME terms, and while have tried to avoid redundancy in our choice of pathways in the plot, several of these terms will contain overlapping genes. The complete dataset table of results is available on the GitHub repository (https://github.com/hilldr/Hill_HIO_Colonization_2017).

Also, as the plots in B suggest that many genes are included with a very small up- or down-regulation, it would be important to have some handle on not just the significance, but also the average absolute size of the change observed. A second set of graphs, or a supplementary figure with more information would be helpful.

To address this concern, we have generated a plot of the log2-transformed fold-change in gene expression relative to PBS-injected controls for all 7 experimental conditions examined in this set of experiments: live *E. coli* +/- SC-514, heat-killed *E. coli* +/- SC-514, and hypoxic culture +/- SC-514, and PBS + SC-514 (Figure 5—figure supplement 1). The format of the plot is identical to the plot shown in Figure 2, and shows the fold-change in expression of all transcripts measured in a given condition relative to PBS treatment alone (grey) with significantly up- and down-regulated transcripts colored in red and blue, respectively. This figure reveals that the scale of the global transcriptional response varies somewhat between experimental conditions. This may be expected, given, for example, that hypoxic culture would be expected to permeate both epithelial and mesenchymal cells throughout the HIO, whereas the impact of heat-inactivated *E. coli* injected into the HIO lumen may be more limited.

E) In the legend, were pathways with enrichment P-values greater than 0.01 excluded?

This was a typographical error. Pathways with P-values > 0.01 were excluded from the plot and the legend has been amended to fix this.

In Figure 5, it would be important to include a group microinjected with dead E. coli, immediately followed by hypoxia, to conclude that both factors act together to induce b-defensins. For Figure 5, does hBD-2 require a reducing agent for activity as hBD1 does (Nature 469, 419-423)? Typically, these pore-forming AMPs to exert a stronger effect on the rapidly growing bacteria than in the stationary phase, and in fact, your maximum growth rate (i.e. maximum curve gradient) is even higher where the BD-2 is added, suggesting that something in the BD-2 may even permit faster E. coli growth in LB. Death over several hours in late stationary phase may be rather due to accumulation of a toxic metabolite. To control for these effects, it will be important to show growth data with heat-inactivation of the BD-2. To focus on killing in the stationary phase a late stationary-phase culture could be treated with differing concentrations of BD-2 over short time-courses (including the inactivated controls), measuring loss of membrane integrity by Sytox-green uptake by flow cytometry or microscopy. As O-antigens can inhibit AMP function, E. coli K-12 could be included in these experiments.

Following the reviewers’ suggestion, we have performed in vitro *E. coli* experiments utilizing heat-inactivated BD-2 as a control (Figure 5—figure supplement 2). We found that heat inactivation of the recombinant BD-2 peptide completely abrogated the reduction in *E. coli* growth caused by addition of BD-2 to the cultures. Furthermore, we confirmed that this recombinant BD-2 has similar inhibitory activity against *E. coli* K-12, suggesting that the effects of BD-2 are not strain specific. We have also included data on the effect of BD-2 on bacterial carrying capacity derived from the growth curves (Figure 5 and Figure 5—figure supplement 2). Our original hypothesis was that up-regulation of AMP expression in HIOs following *E. coli* microinjection (Figure 2, Figure 5) may contribute to limited *E. coli* growth in the HIO lumen (Figure 1). BD-2 significantly reduces carrying capacity in vitro at concentrations consistent with conditions in the HIO (Figure 5 and Figure 5—figure supplement 2). This demonstrates that an AMP expressed by HIOs in response to *E. coli* colonization has the potential to reduce microbial growth in the HIO. As pointed out by the reviewer, questions remain as to the mechanism of action of human β defensins. Multiple mechanisms of action have been proposed in the literature (Ulm et al. *Front. Immunol*. 2012, Brogden, 2005) including pore formation, transcriptional and metabolic inhibition, inhibition of membrane transporters, etc., which vary between defensin family members and specific bacterial targets and can be influenced by the local biochemical conditions such as redox state (*Nature* 469, 419-423). While we agree that this presents an interesting question, we have not defined the specific mechanism(s) of action for BD-2 against *E. coli* str. ECOR2. Such mechanistic studies involve complex structural biology and bacterial genetics experiments and lie outside of our expertise, the time limitations on this revision, and are somewhat outside the intended scope of this manuscript.

In relation to Figure 6, can induction of mucus production and induction of goblet cell differentiation be delineated? The slow appearance of mucin gene upregulation appears more consistent with a differentiation phenotype than simple gene expression?

This is a question that we have asked ourselves from an early point in this project. To date, we have found no evidence that the increase in mucus production is associated with an increase in the number of goblet cells. As mentioned above, we saw no difference in the proportion of epithelial cells expressing goblet, Paneth, or enteroendocrine cell markers (MUC2, DEFA5, and CHGA, respectively). Based on staining and RNA-seq transcript counts, Mucin 5AC appears to be more abundant than Mucin 2 in *E. coli* colonized HIOs (Figure 7 and Figure 7—figure supplement 1). This may be significant, as Mucin 5AC is not specific to the goblet cell but is widely expressed by the small intestinal enterocytes. The increase in Mucin expression correlates with the emergence of DPPIV+ epithelial cells, suggesting that the mature enterocyte population contributes to the increase in mucus production.

The interpretation that 'Epithelial barrier integrity is enhanced following bacterial association' (p13) is rather at odds with the data in Figure 7 where the PBS and E. coli treated permeability is the same. Is the meaning that NFkappaB signaling is required for the compensatory effects of the barrier in E. coli-treated organoids? What is the effect of the inhibitor alone?

The reviewers are correct to note that *E. coli* colonization itself does not result in a decrease in barrier permeability compared to PBS, and that the presence of NF-κB inhibitor results in increased barrier permeability. This implies that NF-κB is required for the mediating the response to *E. coli* colonization. Our original intention was to highlight two findings: 1) that maintenance of epithelial barrier integrity following *E. coli* colonization is NF-κB dependent, and 2) that bacterial colonization mitigates damage to the epithelial barrier and increases in epithelial barrier permeability following exposure to pro-inflammatory cytokines. We originally highlighted these two findings under a single header, stating that both were evidence of enhanced barrier integrity. The reviewer's comments, however, make it clear that these data are probably best presented as two distinct findings, and that is the approach we took in the Discussion. We have therefore separated our description of these results into two distinct headers in the Results section. We hope that this will clarify and highlight the effects of bacterial colonization on HIO barrier function and improve consistency with the interpretation we have laid out in the Discussion.

We have also added an additional treatment group to Figure 8. HIOs were microinjected with PBS alone 24 h prior to microinjection with FITC-dextran and the addition of HIO media containing SC-514 ('PBS + SC-514'). Treatment with SC-514 alone had no detrimental effect on barrier function. We interpret this data as demonstrating that inhibition of the NF-κB pathway prevents the epithelial response to bacterial stimuli (both bacterial contact and hypoxia) that normally bolsters barrier function. The result is increased epithelial barrier permeability (Figure 8) and a higher incidence of bacterial translocation (Figure 8).